# Anomalies of Non-Invertible Symmetries in (3+1)d

Clay Córdova[1], Po-Shen Hsin[2], and Carolyn Zhang[1,3]

[1]*Department of Physics, Kadanoff Center for Theoretical Physics & Enrico Fermi Institute, University of Chicago*

[2]*Mani L. Bhaumik Institute for Theoretical Physics, Department of Physics and Astronomy, University of California Los Angeles*

[3]*Department of Physics, Harvard University, Cambridge, MA02138, USA*

## Abstract

Anomalies of global symmetries are important tools for understanding the dynamics of quantum systems. We investigate anomalies of non-invertible symmetries in 3+1d using 4+1d bulk topological quantum field theories given by Abelian two-form gauge theories, with a 0-form permutation symmetry. Gauging the 0-form symmetry gives the 4+1d "inflow" symmetry topological field theory for the non-invertible symmetry. We find a two levels of anomalies: (1) the bulk may fail to have an appropriate set of loop excitations which can condense to trivialize the boundary dynamics, and (2) the "Frobenius-Schur indicator" of the non-invertible symmetry (generalizing the Frobenius-Schur indicator of 1+1d fusion categories) may be incompatible with trivial boundary dynamics. As a consequence we derive conditions for non-invertible symmetries in 3+1d to be compatible with symmetric gapped phases, and invertible gapped phases. Along the way, we see that the defects characterizing $\mathbb{Z}_4$ ordinary symmetry host worldvolume theories with time-reversal symmetry $\mathsf{T}$ obeying the algebra $\mathsf{T}^2 = C$ or $\mathsf{T}^2 = (-1)^F C$, with $C$ a unitary charge conjugation symmetry. We classify the anomalies of this symmetry algebra in 2+1d and further use these ideas to construct 2+1d topological orders with non-invertible time-reversal symmetry that permutes anyons. As a concrete realization of our general discussion, we construct new lattice Hamiltonian models in 3+1d with non-invertible symmetry, and constrain their dynamics.

April 23, 2024

# 1  Introduction

Symmetry plays a crucial role in our understanding of quantum systems. In particular, 't Hooft anomalies of global symmetries are invariant across all energy scales, and are powerful tools for constraining dynamics. Examples of anomalies in nature are abundant, including for instance chiral anomalies in gauge theories, Lieb-Schultz-Mattis anomalies for lattice models, as well as examples of anomalies of discrete symmetries.

In recent years, the concept of symmetry has been generalized in various directions (see e.g. [1] for a review with references). In relativistic continuum quantum field theories a working definition of a symmetry is any topological operator of the system. This includes ordinary global symmetries (topological operators of codimension one in spacetime) as well as higher-form symmetries (topological operators of higher codimension) [2]. A particularly novel generalization is to *non-invertible* symmetries, which are symmetries generated by topological operators without inverses. Non-invertible symmetries include the familiar Kramers-Wannier duality, and other topological line defects in 1+1d [3–22]. Beyond 1+1d systems, non-invertible symmetries are also ubiquitous in higher dimensions as discussed in *e.g.* [23–26] in 2+1d, and *e.g.* [26–81] in higher spacetime dimensions. Generalized symmetry also plays a role in the weak gravity conjecture and the completeness hypothesis [37, 82–86], as well as particle physics applications [41, 42, 52, 54, 56, 77, 87, 88].

Non-invertible symmetries can also be anomalous, leading to new constraints on the dynamics of quantum systems. In the case of ordinary symmetries, an anomaly is often defined as an obstruction to gauging the global symmetry, i.e. summing over insertions of the associated topological operators. A consequence of a non-trivial anomaly is then that the system cannot be deformed to a trivially gapped phase by any continuous symmetry preserving deformation including renormalization group flow. For non-invertible symmetries, these two points of view on anomalies may in general differ [89], and below we will directly define anomalies of non-invertible symmetries as obstructions to trivially gapped realizations of the symmetry. Our main results are to characterize certain anomalies of non-invertible symmetries in 3+1d.

Anomalies of non-invertible symmetries in 1+1d can be systematically understood using

fiber functors [13, 15, 90–92]. These anomalies depend on the $F$-symbol, which generalizes the 3-cocycle defining anomalies of invertible symmetries. While systematic, this method has two main drawbacks. First, it provides more information than just presence or absence of an anomaly; it completely defines a trivially gapped phase in the absence of an anomaly. Because fiber functors provide more information than desired, they are also very difficult to use in general. For example, it is very difficult to determine whether or not a fiber functor for a given non-invertible symmetry exists. Second, this approach is difficult to generalize to higher dimensions.

Recently, [28, 38, 39, 65, 66, 93, 94] made progress in understanding anomalies of particular kinds of non-invertible symmetries in 3+1d. However, the framework employed only applies to specific non-invertible symmetries. Moreover, they do not take into account the generalization of the $F$-symbol, which includes in particular the 3+1d analogue of the 1+1d Frobenius-Schur (FS) indicator. We denote this important piece of data defining the symmetry by $\omega$ (or $\omega_f$ for fermionic systems). In general, the FS indicator is one piece of data entering into the higher-categorical structure of the non-invertible symmetry [34, 47, 59, 70, 72, 73, 75, 95].

Below, we provide an alternate approach for detecting whether or not certain kinds of non-invertible symmetries are anomalous. This approach is applicable to non-invertible symmetries that include Kramers-Wannier-like (duality and more general $n$-ality) defects in any spacetime dimension. In 1+1d, this is quite restrictive, but in 3+1d, we will show that this actually encompasses all finite non-invertible symmetries. Our approach refines of the above studies of non-invertible symmetries in 3+1d to include anomalies due to $\omega$. Specifically, [28, 39, 65] showed that for a given kind of gauging, certain 1-form SPTs, labeled by integers $(N, p)$, in 3+1d are invariant and therefore can have duality defects. We show that for trivial $\omega$ or $\omega_f$, the 1-form symmetries defined by those valid $(N, p)$ together with the duality symmetry do indeed form anomaly-free non-invertible symmetries. On the other hand, for nontrivial $\omega$ or $\omega_f$, the symmetry is always anomalous for $N$ odd, but can be anomaly-free or anomalous for $N$ even. Our main results are stated in Theorem 1 and Theorem 2. Our approach also reproduces, via a quicker and easier calculation, the results of [15, 96, 97] for non-invertible symmetries in 1+1d. It furthermore provides an interpretation of the physical meaning of the anomaly.

## 1.1 Symmetry TQFTs for 3+1d non-invertible symmetries

Our approach uses the symmetry topological quantum field theory (TQFT), which is a theory in one higher dimension than the physical system carrying the anomaly. Any QFT can be viewed as a symmetry TQFT with appropriately chosen boundary conditions [2, 48, 49, 98–100], with the symmetry given by bulk defects restricted to the boundaries. More precisely, the boundary of a TQFT is a relative theory [101–103], and we need to further choose a polarization to obtain an absolute theory, without the bulk TQFT. Specifically, we can put the bulk TQFT

on an open interval with one topological boundary corresponding to the choice of polarization, giving the desired symmetry, and the other boundary chosen such that shrinking the interval removes the bulk and produces the QFT of interest [101–103]. From this perspective, constraints from anomalies of the symmetry can be viewed as constraints on the possible boundary dynamics of the given bulk TQFT. Symmetry TQFTs were used in [66, 97, 99] to study anomalies of non-invertible symmetries in 1+1d (and some aspects of non-invertible symmetries in higher dimensions), and have been further explored in [95, 104–107]. From this perspective, we can completely classify finite non-invertible symmetries using TQFTs in one higher dimension that have at least one gapped boundary condition.

Non-invertible symmetries in 1+1d are diverse because 2+1d TQFTs are diverse. However, in 4+1d, TQFTs with bosons are all Witt equivalent (i.e. have topological interfaces) to Abelian two-form gauge theories [108, 109] (possibly with a fermion). This is because all the particles are bosonic, so we can always condense them and the resulting theory is always an Abelian two-form gauge theory. This means that all TQFTs with bosons in 4+1d can be obtained from gauging a 0-form symmetry $G$ of an Abelian two-form gauge theory. Our approach applies to all symmetries for which the symmetry TQFT can be obtained by gauging a 0-form symmetry of an Abelian gauge theory, so in 3+1d, it can in fact be used to study all finite 3+1d non-invertible symmetries.

For concreteness, we will focus on symmetries with duality-like defects, whose symmetry TQFTs are obtained by gauging an Abelian permutation symmetry of an Abelian gauge theory. For example, Tambara-Yamagami fusion category symmetries in 1+1d have symmetry TQFTs given by gauging a $\mathbb{Z}_2$ permutation symmetry of an Abelian 1-form gauge theory [49, 97, 110]. Our main interest lies in Tambara-Yamagami-like symmetries in 3+1d, generated by a 1-form $\mathbb{Z}_N$ symmetry and a non-invertible duality symmetry, whose symmetry TQFT is a $\mathbb{Z}_N$ 2-form gauge theory with a gauged $\mathbb{Z}_4$ permutation symmetry[1]. In general, the theory resulting from gauging a permutation symmetry is rather complicated, with various non-invertible higher-form symmetries. However, its properties are already fully determined by two simpler pieces of data: (1) the 0-form symmetry action on the $\mathbb{Z}_N$ 2-form gauge theory and (2) the choice of SPT of the 0-form $G$ symmetry stacked on the system prior to gauging. We will show that these two pieces of data specify the anomaly: (1) determines whether or not there is a "first level obstruction" like those studied in Refs. [28, 39, 65] and (2) determines a "second level obstruction" related to $\omega$ and $\omega_f$.[2]

---

[1]We give the full fusion rules in Eq. (2.13).

[2]The higher fusion category characterizing the symmetry also depends on fractionalization data, i.e. the possible decoration of junctions of codimension one symmetry defects by codimension two symmetry defects (i.e. the one-form symmetry operators). In general, this data also modifies the anomaly (see e.g. [111–113]). However, in our case we are focused on examples that are self-dual under gauging the one-form global symmetry. This implies that anomalies involving the one-form symmetry are trivial and hence the fractionalization choice does not modify the anomaly of the QFTs of interest.

### 1.1.1   4+1d Abelian 2-form gauge theory

We begin with the first piece of data, which may already indicate that the 3+1d non-invertible symmetry is anomalous. A general Abelian two-form gauge theory is described by the action

$$S = \sum_{I,J} \frac{iK^{IJ}}{4\pi} \int b^I db^J \ , \tag{1.1}$$

where $b^I$ are $U(1)$ two-form gauge fields, and $K$ is an antisymmetric matrix.[3] Our main interest is the simplest example of the above which is a $\mathbb{Z}_N$ 2-form gauge theory,

$$S = \frac{iN}{2\pi} \int b_e db_m \ . \tag{1.2}$$

It is characterized by loop excitations labeled by integers $(q_e, q_m) \in \mathbb{Z}_N \times \mathbb{Z}_N$, with antisymmetric braiding [29,104,109,115]. Different kinds of non-invertible defects in 3+1d correspond to duality symmetries in the 2-form gauge theory with different permutation actions. For example, $S : (b_e, b_m) \to (b_m, -b_e)$ and $ST : (b_e, b_m) \to (b_m, -b_e + b_m)$. As we will discuss more in detail in Section 2, the permutation action of the 0-form symmetry can already indicate an anomaly: if there does not exist a subgroup of $N$ loops that (1) can simultaneously condense, (2) is invariant under the duality symmetry, and (3) overlaps trivially with those generated by $(q_e, q_m) = (1, 0)$, then the 3+1d non-invertible symmetry is anomalous. A collection of loops fulfilling these criteria is the 4+1d analogue of the "duality-invariant magnetic Lagrangian subgroup" described in Ref. [97]; different such 4+1d subgroups correspond to different 3+1d duality-invariant 1-form SPTs. By studying these subgroups, we will reproduce and generalize the results derived in Refs. [28,39,65]. For example, we will rederive the fact that for $S$ gauging, $-1$ must be a quadratic residue mod $N$ for the non-invertible symmetry to be anomaly-free.

### 1.1.2   4+1d SPT/Frobenius-Schur indicator

If the two-form gauge theory has the kind of Lagrangian subgroup described above, the symmetry passes the first level obstruction and we can consider the second piece of data, which can present other anomalies. The second piece of data describes stacking of a 4+1d SPT of the 0-form symmetry on the Abelian gauge theory before gauging. The fusion of the fluxes of the SPT modifies the fusion of the duality defects.[4] In 2+1d, the SPT for a $\mathbb{Z}_2$ duality symmetry is classified by $H^3(\mathbb{Z}_2, U(1))$, and is precisely the Frobenius-Schur (FS) indicator. This quantity affects the $F$-symbol of the duality object of 1+1d Tambara-Yamagami fusion

---

[3]We can also include diagonal entries in $K$, which give rise to fermionic loop excitations [104,114]. For simplicity, we do not consider such cases here.

[4]In Appendix B, we show that $\omega$ is also related to braiding correlation functions for domain wall operators in 4+1d, similar to how the FS indicator in 2+1d is related to the self-statistics of line operators [116].

categories. In 4+1d, stacking with an SPT for the 0-form symmetry gives the analogue of the FS indicator we denoted by $\omega$ above. Because $\omega$ affects the fusion of the duality defects in 3+1d [111,117], it plays an important role in determining the anomaly of the 3+1d non-invertible symmetry.

We will be particularly interested in order four duality symmetries. Then, the relevant $\mathbb{Z}_4$ SPTs are those which directly interplay with the $\mathbb{Z}_4$ symmetry and hence are given by the quotient below

$$\Omega^5_{SO}(B\mathbb{Z}_4)/\Omega^5_{SO}(\text{pt}) \cong \mathbb{Z}_4 \times \mathbb{Z}_4 \ , \quad \Omega^5_{Spin}(B\mathbb{Z}_4)/\Omega^5_{Spin}(\text{pt}) \cong \mathbb{Z}_4 \ , \tag{1.3}$$

where above $\Omega^5(\text{pt})$ denotes purely gravitational SPTs, and the two cases above correspond to respectively bosonic or spin SPTs. Thus, the higher analogs of the FS-indicator, $\omega$ and $\omega_f$ takes values in $\mathbb{Z}_4 \times \mathbb{Z}_4$ and $\mathbb{Z}_4$ respectively. For reasons discussed in Section 3.4, we will consider $\omega$ for $N$ odd and $\omega_f$ for $N$ even. If $\omega$ or $\omega_f$ is trivial, then it does not present any additional anomaly; all we must check is the existence of a duality-invariant magnetic Lagrangian subgroup as described above. If it is nontrivial, then our main strategy is as follows: $\omega$ and $\omega_f$ describe $\mathbb{Z}_4$ SPTs in 4+1d that have a decorated domain wall construction, that we explain in Section 3.4.1 following [65,118,119]. Specifically, $\omega$ describes 4+1d SPTs where the the $\mathbb{Z}_4$ domain walls are decorated by 3+1d SPTs of $\mathsf{T}^2 = C$ symmetry, where $\mathsf{T}$ is time-reversal and $C$ is charge conjugation satisfying $C^2 = 1$, and $\omega_f$ describes 4+1d SPTs where the $\mathbb{Z}_4$ domain walls are decorated by 3+1d $\mathsf{T}^2 = (-1)^F C$ SPTs. Note that this correspondence does not require the ambient 4+1d theory to have time-reversal symmetry; rather $\mathsf{T}$ is a symmetry of the worldvolume theory of the defect. This decorated domain wall construction means that when a $\mathbb{Z}_4$ domain wall ends at the boundary, its 2+1d endpoint hosts a theory with a $\mathsf{T}^2 = C$ or $\mathsf{T}^2 = (-1)^F C$ anomaly. The 3+1d non-invertible symmetry is then anomaly free if and only if the 2+1d duality defects also have a $\mathsf{T}^2 = C$ or $\mathsf{T}^2 = (-1)^F C$ anomaly, that can cancel that of domain wall endpoints. By determining the anomalies of the SPTs on the $\mathbb{Z}_4$ domain walls and the anomalies of the duality defect theories, we find that this cancellation occurs when $N$ even but not for $N$ odd. Furthermore, for even $N$, this cancellation only occurs for even classes of $\omega_f$. Therefore, nontrivial $\omega$ and $\omega_f$ can make the symmetry anomalous, in certain cases.

Our method applies to Kramers-Wannier-like symmetries in general spacetime dimensions. As a warm-up to our main derivations, we reproduce the result that the 1+1d $\mathbb{Z}_2 \times \mathbb{Z}_2$ Tambara-Yamagami fusion category [120] with the diagonal bicharacter and non-trivial FS indicator is anomalous, but that with the off-diagonal bicharacter and nontrivial FS indicator is anomaly-free [15]. Here, the domain walls of the 2+1d $\mathbb{Z}_2$ SPT carry 1+1d $\mathsf{T}$ SPTs, and the duality defect theory is a free qubit.

## 1.2 Anomalies of $\mathsf{T}^2 = (-1)^F C$ and non-invertible $\mathsf{T}$ symmetry

As we show in Section 3.6.1 and Section 3.6.2, the anomalies for $\mathsf{T}^2 = C$ symmetry in 2+1d admit $\mathbb{Z}_4 \times \mathbb{Z}_4$ classification. Similarly, the anomalies of $\mathsf{T}^2 = C(-1)^F$ symmetry in 2+1d admit $\mathbb{Z}_4$ classification. These symmetry algebras involving $\mathsf{T}$ are common in 2+1d [121–123] and as such this classification is of intrinsic interest. The anomalies can be detected as follows. In the bosonic case, the anomalies $\omega = (k, \ell) \in \mathbb{Z}_4 \times \mathbb{Z}_4$ means that the theory has chiral central charge $c_- = 2\ell$ mod 8. The anomaly $k$ can be detected by gauging the $C$ symmetry (unitary $\mathbb{Z}_2$ symmetry in 2+1d is non-anomalous since $H^4(\mathbb{Z}_2, U(1)) = 0$). When $k = 2$, gauging the $C$ symmetry renders the $\mathsf{T}$ symmetry into a 2-group symmetry; when $k$ is odd, gauging the $C$ symmetry makes $\mathsf{T}$ symmetry non-invertible. In the fermionic case, the anomaly $\omega_f \in \mathbb{Z}_4$ implies the theory has chiral central charge $c_- = \omega_f/4$ mod 1/2 for odd $\omega_f$. For $\omega_f = 2$, gauging the $C$ symmetry renders the time-reversal symmetry non-invertible.

In the process of studying $\mathsf{T}^2 = (-1)^F C$ anomalies, we also find various 2+1d systems with non-invertible time-reversal symmetry that are interesting in their own right. Specifically, we find an infinite family of 2+1d TQFTs that have non-invertible time-reversal symmetry, denoted by $O^{N,p}$, that are obtained by gauging a $\mathbb{Z}_2$ unitary charge conjugation symmetry in the minimal Abelian TQFT $\mathcal{A}^{N,p}$ [124] with even $N$ and $p^2 = -1$ mod $N$. The original $\mathcal{A}^{N,p}$ theory has an anomalous $\mathsf{T}^2 = (-1)^F C$ symmetry. As a result, $O^{N,p}$ has an anti-unitary non-invertible symmetry that implements time-reversal transformation composed with coupling to a $\mathbb{Z}_2$ gauge theory.

The fusion rules of the non-invertible time-reversal symmetry generator with its orientation reverse produces a sum including both the identity and (copies of) the Kitaev chain [125, 126]. Other examples of non-invertible time-reversal symmetry in 3+1d were studied in [44].

## 1.3 Lattice models

Finally, we also provide concrete lattice models for 3+1d theories with $\mathbb{Z}_N$ 1-form symmetries, where the matter degrees of freedom live on the edges of a cubic lattice. These lattice models are invariant under $S$ gauging. We conjecture a phase diagram in the space of couplings, $N$, and $p$; the phase diagram has only previously been considered for $p = 0$ [65, 127, 128]. We also consider aspects of $ST^{2n}$ gauging on the lattice, highlighting some subtleties that will be further explored in future work.

**Note Added**   Near the completion of this work, we learned of recent work [94] that also discusses duality-invariant Lagrangian subgroups in 4+1d, as well as [129], which also discusses the anomalies of non-invertible symmetries.

# 2 First level obstruction: Lagrangian subgroups

As mentioned in the introduction, every 4+1d TQFT can be obtained by gauging a 0-form symmetry of an Abelian 2-form gauge theory, possibly with a transparent fermion. Let us first present an intuitive argument for this result (see Refs. [108, 109] for more details). We will show that if we ungauge the 0-form symmetry by condensing all the particles, the resulting theory is an Abelian 2-form gauge theory, possibly with a transparent fermion.[5]

The particles in a fermionic 4+1d TQFT consist of bosons, emergent fermions, and a transparent fermion. These particles can all be simultaneously condensed because the emergent fermions can be paired with the transparent fermion. After condensing the particles, we obtain a theory with only loop excitations. We need to prove that these loop excitations have an Abelian fusion algebra.

Suppose that the fusion of two simple loop excitations $s, s'$ were non-Abelian, *i.e.*

$$s \times s' = \sum_i s_i \ , \tag{2.1}$$

where the right hand is a direct sum of loop excitations. Let us shrink the circumference of the loops so that they become particle excitations.[6] Since there are no non-trivial particles left, we find

$$1 \times 1 = \sum_i 1 \ , \tag{2.2}$$

which gives a contradiction unless the right hand side only contains a single term, *i.e.* the fusion algebra is Abelian. The TQFT after condensing the particles is therefore an Abelian 2-form gauge theory.

Since every bosonic 4+1d TQFT with emergent fermions can be obtained from a fermionic one by gauging fermion parity, every bosonic 4+1d TQFT can also be obtained by gauging a 0-form symmetry of an Abelian 2-form gauge theory, possibly with a transparent fermion.

The 3+1d non-invertible symmetries we are interested in have symmetry TQFTs where the 0-form symmetry acts on the loop excitations by permutation. In this section, we discuss anomalies determined by these permutation actions.

---

[5]Equivalently, every 4+1d TQFT is Witt equivalent to an Abelian 2-form gauge theory, possibly with a transparent fermion. This means that there is a topological interface between the two theories.

[6]The reduction of loop excitations to particle excitations by shrinking is also used in *e.g.* [130] for the classification of 3+1d TQFTs.

## 2.1 Review of Abelian 2-form gauge theory

We begin by reviewing some aspects of Abelian 2-form gauge theories. See Ref. [29] and references therein for more details. Such a theory can be described by the action

$$\sum_{I,J=1}^{2r} \frac{K_{IJ}}{4\pi} \int b^I db^J = \sum_{I<J} \frac{M_{IJ}}{2\pi} \int b^I db^J \ , \tag{2.3}$$

where $K = M - M^T$ is an antisymmetric, non-degenerate matrix, with $I, J = 1, \cdots 2r$. $b^I$ are 2-form $U(1)$ gauge fields, and we will also label them by $b^I = b_e^I$ and $b^{r+I} = b_m^I$ for $I = 1, \cdots r$. Note that in terms of matrix $M$, the action is properly quantized for each term, while each term separately is not properly quantized in the expression with $K$ if $K_{IJ}$ is odd.

The theory consists of Abelian loop excitations, described by surface operators $e^{i \oint q^I b^I} = e^{i \oint (q_e^I b_e^I + q_m^J b_m^J)}$ labeled by integer vectors $q = \{q^I\} = \{q_e^I, q_m^J\}$. Unlike Abelian particle (anyon) excitations of 2+1d TQFTs, the loop excitations $\{q^I\}$ and $\{q'^I\}$ have antisymmetric braiding, given by

$$\langle q, q' \rangle = e^{2\pi i q^T K^{-1} q'} = \langle q', q \rangle^* \ . \tag{2.4}$$

From (2.15), we see that excitations of the form $K^{IJ} q^I$ for any integer vector $q$ are trivial. Therefore, the loops fuse according to an Abelian group given by

$$\mathcal{A} = \mathbb{Z}^{2r} / K \mathbb{Z}^{2r} \ . \tag{2.5}$$

The theory has symmetry $g \in GL(2r, \mathbb{Z})$ that transforms the two-form gauge fields $\{b^I\}$ (and thus the charges $\{q^I\}$) while preserving the braiding $\langle q, q' \rangle$:[7]

$$g^T K^{-1} g = K^{-1} \text{ mod } \mathbb{Z} \ . \tag{2.6}$$

Such transformations include those that satisfy $g'^T K g' = K$, where $g'^T = g^{-1}$. This symmetry group consists of all symmetries that permute the loop excitations, so we call it $\text{Aut}(K)$.

### 2.1.1 Aut($K$) and Lagrangian subgroups

We would like to constrain the dynamics of 3+1d theories with non-invertible symmetry corresponding to $g \in \text{Aut}(K)$. To study the first-level obstruction, we must determine which gapped boundaries of a 2-form gauge theory described by $K$ are compatible with a given $g \in \text{Aut}(K)$.

The gapped boundaries of an Abelian 2-form gauge theory are given by Lagrangian subgroups of $\mathbb{Z}^{2r} / K \mathbb{Z}^{2r}$. They correspond to subgroups of the loop excitations whose condensation completely trivialize the theory. In other words, gauging the 2-form symmetry

---

[7]Similar methods can be used to study symmetries of Abelian Chern-Simons theories [122].

generated by the surface operators in the Lagrangian subgroup turns the theory into the trivial theory, given by action

$$\sum_{I,J=1}^{2r} \frac{\mathcal{J}_{IJ}}{4\pi} b'^I db'^J \; , \tag{2.7}$$

where $\mathcal{J} = \begin{pmatrix} 0 & 1_r \\ -1_r & 0 \end{pmatrix}$. The fields $\{b^I\}$ are related to $\{b'^I\}$ by a $GL(2r, \mathbb{Z})$ transformation $U$: $b'^I = U^{IJ}b^J$, where

$$K = U^T \mathcal{J} U \; . \tag{2.8}$$

Note that $U$ does not have determinant $\pm 1$, because it is not simply a change of basis for the fields; in general, $\det(U) = \sqrt{\det(K)}$. This can also be understood from the fact that we have trivialized the theory by gauging a 2-form symmetry leading to new well-defined gauge fields $b'^I$. The matrix $U$ also specifies a subgroup of the loop excitations $e^{i \oint q'^I b'^I}$ that we can simultaneously condense because they all have trivial braiding with each other, according to (2.15). We can denote the subgroup formed by these condensed loops by $\Lambda(U)$:

$$\Lambda(U) = (\mathrm{Im}\, U|_{\mathbb{Z}^{2r}})/K\mathbb{Z}^{2r} \subset \mathbb{Z}^{2r}/K\mathbb{Z}^{2r} \; . \tag{2.9}$$

For the domain wall $g$ to end on the corresponding gapped boundary, we demand that $U$ to commute with $g$

$$UgU^{-1}g^{-1} = 1 \; . \tag{2.10}$$

This means that $\Lambda(U)$ is invariant under $g$.

### 2.1.2 Polarization for the boundary theory: fixing the symmetry

In addition to the constraint above that there exists a $g$-invariant Lagrangian subgroup, there is one further constraint that must be satisfied related to this first-level obstruction. This constraint is that the the Lagrangian subgroup must intersect trivially with a canonical Lagrangian subgroup specified by the polarization. Recall that the 3+1d non-invertible symmetries of interest consist of non-invertible defects together with a finite, Abelian 1-form symmetry. The polarization that gives this 1-form symmetry in the symmetry TQFT corresponds to a Lagrangian subgroup $\Lambda_e$ of the 4+1d Abelian 2-form gauge theory consisting of loops labeled by $q = \{q_E^I, 0\}$. Therefore, in order to pass the first-level obstruction to the symmetry being anomaly-free, we must ensure that there exists a Lagrangian subgroup that is not only invariant under $g$, but also intersects trivially with $\Lambda_e$:

$$\Lambda_e \cap \Lambda(U) = \{0\} \tag{2.11}$$

This condition is the generalization of the existence of duality-invariant *magnetic* Lagrangian subgroup in the study of 1+1d Tambara-Yamagami fusion category symmetries [97].

Lagrangian subgroups that have nontrivial intersections with $\Lambda_e$ give, in the open interval setup with $\Lambda_e$ condensed on one boundary and $\Lambda(U)$ condensed on the other boundary, TQFTs with deconfined particle excitations. The deconfined particle excitations are precisely those in the overlap $\Lambda_e \cap \Lambda(U)$. Such 3+1d theories are not invertible, and generally have nontrivial ground state degeneracy on manifolds other than $S^4$.

**Symmetry-enforced gaplessness** If we are simply interested in symmetric TQFTs rather than symmetric invertible TQFTs, then we can drop the requirement (2.11). Lagrangian subgroups satisfying (2.10) but not (2.11) describe nontrivial 3+1d TQFTs that are invariant under gauging. When a symmetry has an anomaly that prevents even a symmetric gapped phase, it demonstrates "symmetry-enforced gaplessness" as discussed in Refs [131, 132] for invertible symmetries. Note that in 1+1d, a symmetric TQFT must be invertible; only in higher dimensions can a symmetry-preserving TQFT be non-invertible. We will give some examples of $\Lambda(U)$ satisfying (2.10) but not (2.11) in Section 2.2.1 and remark on the effect of $\omega$ and $\omega_f$ on these theories in Section 3.4.

## 2.2 Kramers-Wannier non-invertible symmetry and generalizations

We will now illustrate the above general discussion in the particular case where the symmetry of the 3+1d theory includes a non-anomalous $\mathbb{Z}_N$ 1-form symmetry, together with duality defects that implement gauging of the 1-form symmetry. In 3+1d, different ways to gauge a $\mathbb{Z}_N$ 1-form symmetry correspond to stacking with different SPTs of the 1-form symmetry before gauging. In terms of the partition function, this amounts to adding a topological action for the 2-form gauge field, which is an quadratic action with coefficient labelled by an integer $n$:

$$ST^n \text{ gauging}: \quad Z[B] \to \sum_b Z[b] e^{\frac{2\pi i}{N} \int bB + \frac{2\pi i n}{2N} \int \mathcal{P}(b)} , \tag{2.12}$$

where $B$ and $b$ are classical and dynamical 2-form gauge fields for the 1-form symmetry respectively, and $\mathcal{P}$ is the generalized Pontryagin square operation (see Eq. 3.31). Invariance under $ST^n$ gauging means that the partition functions on the two sides are equal. We will study the first-level obstruction described above for non-invertible defects corresponding to general $ST^n$ gauging. For the special case $n = 0$, which was previously studied in Refs. [26, 28, 39, 49, 66], the non-invertible symmetry consists of the $\mathbb{Z}_N$ 1-form symmetry together with the duality defect $\mathcal{D}$, the charge conjugation defect $\mathcal{U}$, and the condensation

defect $\mathcal{C}_0$, obeying the following fusion rules (specifically for $n = 0$):

$$\bar{\mathcal{D}} \times \mathcal{D} = \mathcal{D} \times \bar{\mathcal{D}} = \mathcal{C}_0$$
$$\mathcal{D} \times \mathcal{D} = \mathcal{U} \times \mathcal{C}_0 = \mathcal{C}_0 \times \mathcal{U}$$
$$\mathcal{U} \times \mathcal{D} = \mathcal{D} \times \mathcal{U} = \bar{\mathcal{D}} \qquad (2.13)$$
$$\mathcal{D} \times \mathcal{C}_0 = \mathcal{C}_0 \times \mathcal{D} = (\mathscr{Z}_N)_0 \, \mathcal{D}$$
$$\mathcal{C}_0 \times \mathcal{C}_0 = (\mathscr{Z}_N)_0 \, \mathcal{C}_0$$

The more general defects studied here, implementing $ST^n$ gauging, obey similar but different fusion rules. For example, the duality defect $\mathcal{D}$ would not be order four in general.

We will constrain the dynamics of theories with defects implementing $ST^n$ gauging using Lagrangian subgroups of 4+1d $\mathbb{Z}_N$ 2-form gauge theories. A $\mathbb{Z}_N$ 2-form gauge theory is described by the Lagrangian

$$\frac{N}{2\pi} b_e db_m, \qquad (2.14)$$

for two-form gauge fields $b_e, b_m$. The gauge field $b_m$ constrains $b_e$ to have $\mathbb{Z}_N$ holonomy, and similarly $b_e$ constrains $b_m$ to have $\mathbb{Z}_N$ holonomy. The theory has loop excitations described by $e^{iq_e \oint b_e + iq_m \oint b_m}$ for integers $\{q_e, q_m\} \in \mathbb{Z}_N \times \mathbb{Z}_N$, that generate a $\mathbb{Z}_N \times \mathbb{Z}_N$ fusion algebra. The excitations $\{q_e, q_m\}$ and $\{q'_e, q'_m\}$ have antisymmetric braiding [29], given by

$$\langle \{q_e, q_m\}, \{q'_e, q'_m\} \rangle = e^{\frac{2\pi i}{N}(q_e q'_m - q_m q'_e)} = \langle \{q'_e, q'_m\}, \{q_e, q_m\} \rangle^* . \qquad (2.15)$$

The theory (2.14) has $SL(2, \mathbb{Z})$ symmetry that transforms the fields $b_e, b_m$ [2, 29]. Domain walls that generate $ST^n \in SL(2, \mathbb{Z})$ in the bulk, when ending on the boundary, become boundary topological defects between theories related by gauging the $\mathbb{Z}_N$ 1-form symmetry with local counterterm $n$. In terms of the partition function $Z[B]$, the two sides are related as in equation (2.12).

### 2.2.1 Gapped boundaries with $ST^n$ non-invertible symmetry

We will consider in this section gapped boundaries of the $\mathbb{Z}_N$ gauge theory that describe duality-invariant TQFTs. These are boundaries where the $ST^n$ domain wall can end, with the same theory on either side of the defect. In Section 2.2.2, we will specify a polarization and restrict to invertible TQFTs by requiring (2.11).

As discussed in Section 2.1.1, the gapped boundaries of an Abelian 2-form gauge theory are labeled by Lagrangian subgroups. The Lagrangian condition means that gauging a 2-form symmetry, which can be expressed as a change of variables $\{b_e, b_m\} \to \{b'_e, b'_m\}$ by a $GL(2, \mathbb{Z})$ transformation $U$, can bring the theory to the trivial theory. For $\mathbb{Z}_N$ gauge theory, the trivial theory is given by

$$\frac{1}{2\pi} b'_e db'_m . \qquad (2.16)$$

The symmetry transformation $g = ST^n \in \mathrm{Aut}(K)$ is given by $\begin{pmatrix} 0 & 1 \\ -1 & n \end{pmatrix}$, which maps

$$b_e \to -b_m \qquad b_m \to b_e + nb_m. \tag{2.17}$$

$U$ must commute with $g$ according to (2.10), so $U$ must take the form

$$U(n, \alpha, \beta) = \begin{pmatrix} \alpha & \beta \\ -\beta & \alpha + n\beta \end{pmatrix}, \tag{2.18}$$

where $\alpha$ and $\beta$ are integers. Substituting the transformation into (2.8) gives

$$N = \alpha^2 + \beta^2 + n\alpha\beta . \tag{2.19}$$

For values of $N$ with solutions to (2.19), there exists duality-invariant Lagrangian subgroups, so the $ST^n$ domain wall can end on the boundary with the same theory on either side. The non-invertible symmetry can therefore be realized in a symmetric TQFT. For other values of $N$, the $ST^n$ defects can only appear in gapless phases.

The subgroup of bulk loop excitations that condenses on the gapped boundary is given by $(\mathrm{Im}\, U(n, \alpha, \beta)|_{\mathbb{Z}^2})/K\mathbb{Z}^2$, since all excitations $\int b'_e, \int b'_m$ are trivial. The Lagrangian subgroup $\Lambda(U(n, \alpha, \beta))$ is therefore generated by loops $\{q_e, q_m\} = \{\alpha, -\beta\}, \{\beta, \alpha + n\beta\}$.

Let us give some examples of $N$ with solutions to (2.19), for a given $n$:

- For $n = 0$, we have

$$N = 2, 4, 5, 8, 9, 10, 13, 16, 17, 18, 20, 25, 26, 29, 32, 34, 36, 37, 40, 41, 45, 49, 50, 52, 53, 58,$$
$$61, 64, 65, 68, 72, 73, 74, 80, 85, 89, 98, 100, 113, 128, \cdots .$$
$$\tag{2.20}$$

  Note that the above $N$ agrees with the entries in Table 1 of [65]. Examples of theories with these defects are $\mathbb{Z}_N$ 3+1d Toric code in a transverse field [28].

- For $n = 1$, we have

$$N = 3, 4, 7, 9, 12, 13, 16, 19, 21, 25, 27, 28, 31, 36, 37, 39, 43, 48, 49, 52, 57, 61,$$
$$63, 64, 67, 73, 75, 76, 79, 84, 91, 93, 97, 108, 109, 112, 127, \cdots . \tag{2.21}$$

- For $n = 2$, we have

$$N = 4, 9, 16, 25, 36, 49, 64, 81, 100, 121, 144, 169, 196, 225, 256, 289, 324, \cdots . \tag{2.22}$$

- For $n = 3$, we have

$$N = 4, 5, 9, 11, 16, 19, 20, 25, 29, 31, 36, 41, 44, 45, 49, 55, 59, 61, 64, 71, 76, 79, 80, 81, 89,$$
$$95, 99, 100, 101, 109, 116, 121, \cdots .$$
$$\tag{2.23}$$

### 2.2.2 Invertible boundaries with $ST^n$ non-invertible symmetry

We now specify the polarization, to study *invertible* duality-invariant 3+1d theories. We choose the polarization to be given by the "electric" Lagrangian subgroup $\Lambda_e$, generated by the $\{1,0\}$ loop. To get an symmetric invertible theory, we must impose (2.11). This means that $\Lambda(U(n,\alpha,\beta))$, generated by loops $\{\alpha,-\beta\},\{\beta,\alpha+n\beta\}$ cannot generate $r\{1,0\}$ for any integer $r \neq 0 \mod N$:

$$\text{Invertible boundary:} \quad \forall p,q \in \mathbb{Z}, \quad p(\alpha,-\beta) + q(\beta,\alpha+n\beta) \neq r(1,0) , \qquad (2.24)$$

for some integer $r \neq 0 \mod N$. Note that $-p\beta + q(\alpha + n\beta) = 0$ can be satisfied by $p = (\alpha + n\beta)m/\ell$ and $q = \beta m/\ell$ for some integer $m$, and $\ell = \gcd(\alpha + n\beta, \beta)$. However, $p\alpha + q\beta = Nm/\ell$, so (2.24) is equivalent to

$$\gcd(\alpha + n\beta, \beta) = 1 . \qquad (2.25)$$

Notice that this means that $\{\alpha, -\beta\}$ generates $\{\beta, \alpha + n\beta\}$ and vice versa, so these two loops are not independent generators.[8] Let us list some examples of theories satisfying (2.25), for a given $n$:

- For $n = 0$, the Lagrangian subgroup is generated by $(\alpha, -\beta)$ and $(\beta, \alpha)$. The first few cases of $N$ with invertible absolute boundaries, labeled by $N_{\alpha,\beta}$, are

$$N = 2_{1,1}, 5_{2,1}, 10_{1,3}, 13_{2,3}, 17_{1,4}, 25_{3,4}, 26_{1,5}, 29_{2,5}, 34_{3,5}, 37_{1,6}, 41_{4,5}, 50_{1,7}, 53_{2,7}, 58_{3,7}, 61_{5,6} \cdots .$$
$$(2.26)$$

  Note that the $N$ above coincide with the entries for SPT in table 1 of [65].

- For $n = 1$, the Lagrangian subgroup is generated by $(\alpha, -\beta)$ and $(\beta, \alpha + \beta)$. The first few cases of $N$ with invertible absolute boundaries are

$$N = 3_{1,1}, 7_{1,2}, 13_{1,3}, 19_{2,3}, 21_{1,4}, 31_{1,5}, 37_{3,4}, 39_{2,5}, \cdots . \qquad (2.27)$$

**Example: bulk and boundary field theories for $N = 5$**   To illustrate the confinement of particles in a concrete example, let us consider $n = 0$ and $N = 5$. The boundary and bulk action is given by

$$\frac{10}{4\pi} \int_{4d} b_m b_m + \frac{5}{2\pi} \int_{5d} b_e db_m . \qquad (2.28)$$

The equation of motion for $b_m$ gives $b_e + 2b_m = 0$ on the boundary, so $\{q_e, q_m\} = \{1, 2\} = 2\{3, 1\}$ is condensed on the boundary, indicating $\alpha = 2, \beta = 1$. To consider an absolute

---

[8]In more detail, Bézout's identity means that there exists integers $x, y$ such that $x\beta + y(\alpha + n\beta) = 0 \mod N$. But if $-x\alpha + y\beta \neq 0 \mod N$, then the Lagrangian subgroup overlaps nontrivially with $\Lambda_e$. Therefore we must have $x\{\alpha, -\beta\} = y\{\beta, \alpha + n\beta\} \mod N$ for some $x, y$.

boundary theory, we can choose $e$-condensed polarization $b_e = da$. The 3+1d absolute theory is then given by

$$\int_{4d} \left( \frac{10}{4\pi} b_m b_m + \frac{5}{2\pi} b_m da \right) \ . \tag{2.29}$$

In this absolute theory, the gauge invariant operators are generated by the open surface operator $e^{i \oint a + 2 \int b_m}$, and there are not genuine line operators [2, 124, 133]. Thus all particles are confined. The 3+1d absolute theory is therefore an invertible TQFT that realizes the non-invertible symmetry.

Note that the gauge invariant open surface operators can be obtained from the loops condensed on the boundary. Thus when the Lagrangian subgroup $\Lambda(U(n, \alpha, \beta))$ intersects trivially with the Lagrangian subgroup of the polarization, all particles are confined.

# 3  Second level obstruction: generalized FS indicators

For the symmetries that pass the first-level obstruction discussed in the previous section, we can consider an additional piece of data denoted by $\omega$ (for odd $N$) or $\omega_f$ (for even $N$). $\omega$ and $\omega_f$ are the generalization of the FS indicator of 1+1d fusion categories and 2+1d TQFTs. In 2+1d TQFTs, the FS indicator can be defined for self-dual anyons satisfying $a \times a \supset 1$. In the case where the self-dual anyons arise from gauging a 0-form $\mathbb{Z}_2$ symmetry the FS indicator comes from stacking with a $\mathbb{Z}_2$ SPT before gauging. The FS indicator therefore modifies the topological spins of the self-dual anyons, and leads to 1+1d boundary fusion category symmetries with different $F$ symbols [110].

We define $\omega$ (or $\omega_f$, for fermionic systems) as the analogous quantity for 3+1d non-invertible symmetries and 4+1d TQFTs. Surface excitations in 4+1d that obey the fusion rule $a^N \supset 1$ come from gauging a $\mathbb{Z}_N$ 0-form symmetry, and we can always stack a $\mathbb{Z}_N$ SPT on the theory before gauging. $\omega$ specifies this SPT, and modifies the braiding correlation functions of the surface excitations (see Appendix B). It therefore partially defines the associator of symmetry defects in the 3+1d boundary theory. For example, for a 3+1d noninvertible symmetry with fusion rule given by (2.13), the 0-form bulk permutation symmetry is $\mathbb{Z}_4$. Therefore, we consider $\omega$ labeling 4+1d $\mathbb{Z}_4$ SPTs, classified by $\mathbb{Z}_4 \times \mathbb{Z}_4$ (in the bosonic case) or $\mathbb{Z}_4$ (in the fermionic case, if the $\mathbb{Z}_2$ subgroup is not identified with $\mathbb{Z}_2^f$). More generally, one must consider SPTs classified by $\Omega_{SO}^5(BG)/\Omega_{SO}^5(\text{pt})$ or $\Omega_{Spin}^5(BG)/\Omega_{Spin}^5(\text{pt})$ [134], because we do not include SPTs that do not involve the $G$ symmetry. The FS indicator $\omega$ or $\omega_f$ can make the 3+1d symmetry anomalous even if it passes the first level obstruction. In this section, we will study anomalies related to $\omega$ and $\omega_f$ using a method applicable to the case where the 4+1d SPT has a decorated domain wall description. Our strategy can be generalized to include more general 4+1d SPTs if we incorporate suitable tangential structures on the domain wall to specify lower dimensional junctions.

## 3.1 Strategy: decorated domain walls and anomaly cancellation

The SPTs labeled by $\omega$ and $\omega_f$ are characterized by the property that domain walls of the bulk 0-form symmetry are decorated with 3+1d SPT phases. When such a domain wall ends on the boundary of the 4+1d bulk, the boundary defect carries the anomaly of the 3+1d SPT. We can cancel the anomaly by decorating the boundary defect with a TQFT, at the cost of modifying the fusion rules of the boundary defects and thereby extending the symmetry. Therefore, the 3+1d non-invertible symmetry is anomaly-free if and only if the fusion rules of the non-invertible defects are compatible with the TQFT that decorates the defects to trivialize the anomaly $\omega$ or $\omega_f$, i.e. it is precisely one of these non-invertible extensions. We will show that in particular the anomalies described by the even classes of $\omega_f$, for $N$ even, can be canceled by an extension to a non-invertible symmetry.

If a symmetry passes the first level obstruction, we then proceed as follows:

1. Determine the 3+1d domain wall SPT given by $\omega$ (for odd $N$) or $\omega_f$ (for even $N$).

2. Check if the TQFT of the duality defect has an anomaly that can cancel that of $\omega$ (for odd $N$) or $\omega_f$ (for even $N$).

Note that if $\omega$ or $\omega_f$ is trivial, then we do not have to go through these steps; only the first level obstruction is relevant. In the following, we will first work out the steps above for 1+1d $\mathbb{Z}_2 \times \mathbb{Z}_2$ Tambara-Yamagami fusion categories, recovering the results from Refs. [15, 96, 97]. We will then study obstructions related to $\omega$ and $\omega_f$ for 3+1d $\mathbb{Z}_N$ Kramers-Wannier symmetries with fusion rules given by (2.13), to both symmetric TQFTs and symmetric invertible TQFTs.

Note that non-invertible symmetries in 3+1d with non-trivial $\omega$ occur in many gauge theories with fermions. An example of such a symmetry is the non-invertible chiral symmetry in quantum electrodynamics [41, 42]. In an upcoming work, we will investigate constraints from these non-invertible symmetries on the dynamics of various gauge theories in 3+1d.

## 3.2 Trivializing the anomaly by symmetry extension in 1+1d

The FS indicator for 1+1d non-invertible symmetries comes from the 2+1d bosonic $\mathbb{Z}_2$ SPT [135], which has the effective action

$$\pi \int A \cup A \cup A \,, \tag{3.1}$$

where $A$ is a background gauge field for the $\mathbb{Z}_2$ symmetry. The anomaly implies that the line defect $\mathcal{N}$ that generates the symmetry in 1+1d is attached to the 1+1d topological action

$$\pi \int A \cup A \,. \tag{3.2}$$

Identifying the $\mathbb{Z}_2$ gauge field $A$ with the first Stiefel-Whitney class $w_1$ of the normal bundle of the domain wall that generates the bulk symmetry, we obtain the action of the 1+1d

bosonic time-reversal ($\mathsf{T}$) SPT, which has defects carrying Kramers doublets [118,119]. Note in particular that the bulk does not in general have $\mathsf{T}$ symmetry even though the defect worldvolume theory does.

To cancel the $\mathsf{T}^2 = -1$ anomaly of these defects, we need to modify the domain walls to cancel the 1+1d SPT phase. These modifications come at the cost of extending the symmetry. There are multiple different ways to extend the symmetry to trivialize the anomaly:

- Invertible extension: decorate the defects with $\pi \int \tilde{a}/2$ where $a$ has the same transformation as $A$ and we pick a lift to $\mathbb{Z}_4$. Then the symmetry becomes a $\mathbb{Z}_4$ symmetry, because the line squares to $(-1)^{\int a}$:

$$\mathcal{N}^2 = (-1)^{\int a}, \quad \mathcal{N}^4 = 1 . \tag{3.3}$$

- Non-invertible extension: decorate the defects with the gapped boundary of $\pi \int a_1 \cup a_2$, where $a_1$ and $a_2$ are $\mathbb{Z}_2$ gauge fields with transformation correlated with $A$. For such a topological surface to end on the defect, we will take the $\mathsf{T}$ symmetry of the domain wall to not permute the Wilson lines $(-1)^{\int a_1}$ and $(-1)^{\int a_2}$ (we will expand on this point in the next section). Then using the method in Refs. [29, 35], the line fuses with itself to produce

$$\mathcal{N} \times \mathcal{N} = 1 + (-1)^{\int a_1} + (-1)^{\int a_2} + (-1)^{\int (a_1 + a_2)} , \tag{3.4}$$

which is the fusion rule of the $\mathbb{Z}_2 \times \mathbb{Z}_2$ Tambara-Yamagami fusion category [120]. Intuitively, the degenerate boundary theory of $\pi \int a_1 \cup a_2$ can absorb the Wilson lines. The fact that the anomaly can be trivialized by the above non-invertible extension is consistent with the fact that the $\mathbb{Z}_2 \times \mathbb{Z}_2$ Tambara-Yamagami fusion category with the off-diagonal bicharacter is anomaly-free even with the nontrivial FS indicator [15, 96, 97].

We remark that decoration of TQFTs on the symmetry generator to cancel the anomaly is also discussed in [27] in the context of gauging a subgroup of anomalous symmetry.

## 3.3 $\mathbb{Z}_2 \times \mathbb{Z}_2$ Tambara-Yamagami symmetries in 1+1d

We will now study in more detail the example of trivializing the above anomaly via non-invertible extensions. In 1+1d, there are four kinds of $\mathbb{Z}_2 \times \mathbb{Z}_2$ Tambara-Yamagami fusion categories with the same fusion rules. They differ in their FS indicator and bicharacter, which together specify the $F$ symbol. We will show that the the $\mathbb{Z}_2 \times \mathbb{Z}_2$ Tambara-Yamagami fusion category with off-diagonal bicharacter can cancel the $\mathsf{T}^2 = -1$ anomaly (as mentioned in the previous section), but the one with diagonal bicharacter cannot. The fusion category with the nontrivial FS indicator and the diagonal bicharacter is therefore anomalous.

The quantum mechanics on the non-invertible line defect can be described by the $\mathbb{Z}_2$ scalars $\phi_1, \phi_2$

$$\pi \int \phi_1 \cup d\phi_2 , \tag{3.5}$$

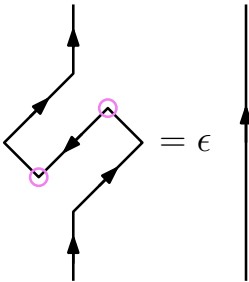

Figure 1: Two time-reversal defects in a duality domain wall, circled in pink, fuse to the FS indicator $\epsilon = \pm 1$. This means that the nontrivial FS indicator, given by $\epsilon = -1$, corresponds to $\mathsf{T}^2 = -1$ on defects.

where $\phi_1, \phi_2$ transform under $\mathbb{Z}_2 \times \mathbb{Z}_2$ unitary symmetry, whose Wilson lines generate the invertible $\mathbb{Z}_2 \times \mathbb{Z}_2$ symmetry. Because this defect is attached to a time-reversal invariant domain wall [118, 119, 126], there is an action of time-reversal on this quantum mechanics . The two bicharacters correspond to two choices of time-reversal action:

$$
\begin{aligned}
\text{Off-diagonal}: \quad & \mathsf{T}(\phi_1) = \phi_1, \quad \mathsf{T}(\phi_2) = \phi_2 \\
\text{Diagonal}: \quad & \mathsf{T}'(\phi_1) = \phi_2, \quad \mathsf{T}'(\phi_2) = \phi_1 .
\end{aligned} \tag{3.6}
$$

These symmetry actions are precisely the involution corresponding to the off-diagonal and diagonal bicharacters respectively (see Ref. [15] for the definition of the involution in terms of the bicharacter). We will call the latter electromagnetic duality symmetry in the quantum mechanics system.[9]

The FS indicator corresponds to the anomaly of the quantum mechanics, described by $\pi \int w_1^2$, which decorates the domain walls of the 2+1d invertible phase $\pi \int A^3$ as described above. From the anomaly, we can see that a nontrivial value of the FS indicator means that $\mathsf{T}^2 = -1$ on the quantum mechanics. Another way to see this directly from the $F$ symbol of the fusion category is illustrated in Fig. 1.

If the fusion category symmetry can be realized by an invertible phase, then the quantum mechanics is well-defined by itself. Clearly, this is the case if the FS indicator is trivial. When the FS indicator is non-trivial, the fusion category symmetry can be realized by an invertible phase if and only if the quantum mechanics can realize the anomaly $\pi \int w_1^2$, *i.e.* if the Hilbert space is in the Kramers doublet projective representation of the time-reversal symmetry.

It is instructive to present the quantum mechanics as a free qubit, where the $\mathbb{Z}_2 \times \mathbb{Z}_2$ symmetry is generated by the Pauli $Z$ and $X$ operators. A non-anomalous time-reversal

---

[9]Note that the choice of involution can also be derived from the permutation action of the $\mathbb{Z}_2$ duality symmetry in the corresponding 2+1d $\mathbb{Z}_2 \times \mathbb{Z}_2$ gauge theory, and applying the $\mathbb{Z}_2$ symmetry along with $\mathcal{CPT}$ as described in Refs. [118, 119].

symmetry can be realized in both cases, with

$$\text{Off-diagonal}: \quad \mathsf{T} = K$$
$$\text{Diagonal}: \quad \mathsf{T}' = HK \; , \tag{3.7}$$

where $K$ is complex conjugation and $H$ is the Hadamard gate.[10] The Hadamard gate satisfies $H^2 = 1$ and $HZH = X$. Therefore, it implements the electromagnetic duality permutation. Both $\mathsf{T}$ and $\mathsf{T}'$ square to the identity, so the above time-reversal symmetries are non-anomalous (*i.e.* the Hilbert space is in a Kramers singlet). Since both cases can realize non-anomalous time-reversal, $\mathbb{Z}_2 \times \mathbb{Z}_2$ Tambara-Yamagami fusion category with trivial FS indicator is anomaly-free, for both the diagonal and off-diagonal bicharacters [96].

### 3.3.1 Off-diagonal bicharacter

Let us couple the quantum mechanics to $\mathbb{Z}_2$ gauge fields $A_1, A_2$, such that $d\phi_1 = A_1, d\phi_2 = A_2$. Then the quantum mechanics has the anomaly

$$\pi \int A_1 \cup A_2 \; . \tag{3.8}$$

In the off-diagonal bicharacter case, since $\mathsf{T}$ does not permute the $\phi_1, \phi_2$ fields, the anomaly remains the same as above in the presence of background $w_1$, and we can choose a "symmetry fractionalization" $A_1 = A_2 = w_1$. This produces the anomaly

$$\pi \int w_1^2 \; . \tag{3.9}$$

We conclude that the fusion category with the off-diagonal bicharacter is anomaly-free, even with a nontrivial FS indicator, in agreement with Ref. [96].

We can also present the "fractionalization" in terms of the free qubit. This means that the time reversal action is correlated with the action of the two $\mathbb{Z}_2$ symmetries. The product of the two $\mathbb{Z}_2$ generators $Z$ and $X$ is the Pauli $Y$ operator, so we obtain

$$\mathsf{T}_{\text{anom}} = YK. \tag{3.10}$$

Since $KYK = -Y$, we find $\mathsf{T}_{\text{anom}}^2 = -1$, *i.e.* the Hilbert space is in a Kramers doublet projective representation. This anomaly cancels that of the FS indicator, recovering the fact that this fusion category is anomaly-free even with a nontrivial FS indicator [96].

---

[10]In the basis of $Z$-eigenvectors, $H = \frac{1}{\sqrt{2}} \begin{pmatrix} 1 & 1 \\ 1 & -1 \end{pmatrix}$

### 3.3.2 Diagonal bicharacter

Let us use the free qubit presentation. In this case, if we try to change the "fractionalization" by correlating the time reversal action with those of the two $\mathbb{Z}_2$ symmetries, we get

$$\mathsf{T}'_{\text{anom}} = YHK \ . \tag{3.11}$$

Using the commutation relation $HYH = -Y$, we find

$$\mathsf{T}'^2_{\text{new}} = YH(-Y)H = Y^2 = +1 \ . \tag{3.12}$$

Thus the Hilbert space is in a Kramers singlet, and there is no anomaly for the time-reversal symmetry. In fact, this is the only consistent way to modify the time-reversal symmetry while preserving the electromagnetic duality permutation.

   We conclude that the quantum mechanics with the $\mathsf{T}$ action given by the diagonal bicharacter cannot cancel the anomaly of the non-trivial FS indicator. This is consistent with the property that the $\mathbb{Z}_2 \times \mathbb{Z}_2$ Tambara-Yamagami fusion category with the diagonal bicharacter is anomalous when the FS indicator is nontrivial [96].

## 3.4   Trivializing the anomaly by symmetry extension in 3+1d

We denote the analogue of the FS indicators in 3+1d by $\omega$ and $\omega_f$ for bosonic and fermionic theories respectively. In this section, we will only consider $S$ gauging, corresponding to 3+1d non-invertible symmetries with fusion rules described by (2.13). Our main results are summarized in Theorem 1 and Theorem 2.

   The duality defect is order four in this case, so $\omega$ and $\omega_f$ label 4+1d bosonic and fermionic $\mathbb{Z}_4$ SPTs respectively. Let us first discuss the domain walls of these SPTs and the 3+1d SPTs that they carry. These 3+1d SPTs correspond to 2+1d anomalies, that can be cancelled by extension in various ways (see Appendix C). Here we will focus on non-invertible extension.

### 3.4.1   4+1d $\mathbb{Z}_4$ SPTs

The 4+1d topological action for the $\mathbb{Z}_4$ gauge field given by the Chern-Simons term will be relevant for both bosonic and fermionic $\mathbb{Z}_4$ SPTs. We will first show that this action can be defined on un-orientable manifolds, but requires a $\mathrm{Wu}_3$ structure. The same structure is present on the domain wall, and is crucial for certain anomalies to be well-defined. The domain wall depends on the $\mathrm{Wu}_3$ structure since it generates $\mathbb{Z}_4$ symmetry, but it has time-reversal symmetry and thus the domain wall can be un-orientable [65].

   Let us begin with the 4+1d Chern-Simons action

$$\frac{2\pi k}{4} \int A \frac{dA}{4} \frac{dA}{4} \ , \tag{3.13}$$

where $A$ is $\mathbb{Z}_4$ gauge field normalized to have integer holonomy $0, 1, 2, 3 \mod 4$. $k = 0, 1, 2, 3$ $\mod 4$ is an integer labeling the $\mathbb{Z}_4$ classification of these SPTs. To examine whether or not we can define this action for $k$ odd on general five manifolds, let us extend the action to a 6d non-orientable bulk manifold (setting here $k = 1$):

$$\pi \int \frac{dA}{2} \frac{dA}{4} \frac{dA}{4} = \pi \int Sq^1 \left( A Sq^2 \left( \frac{dA}{4} \right) \right) \tag{3.14}$$

where $Sq^i$ are the Steenrod squares. Further simplifying using the Cartan formula and the Adem relations,[11] we obtain

$$\pi \int \frac{dA}{2} \frac{dA}{4} \frac{dA}{4} = \pi \int Sq^1 Sq^2 A \frac{dA}{4} = \pi \int Sq^3 A \frac{dA}{4} = \pi \int w_1 w_2 A \frac{dA}{4} , \tag{3.15}$$

where the last relation comes from $Sq^{d-i} a = v_i a$ on a $d$ dimensional manifold. Here, $i$ is the degree of $a$ and $v_i$ is $i$th Wu class. In particular, $v_3 = w_1 w_2$. Thus we can define the action on a general manifold by introducing the coupling

$$\frac{2\pi}{4} \int A \frac{dA}{4} \frac{dA}{4} + \pi \int \rho A \frac{dA}{4} , \tag{3.16}$$

where $d\rho = w_1 w_2$ is the Wu$_3$ structure. Such a structure exists in any closed manifolds of dimension below or equal to five. Under a time-reversal transformation, $\rho \to \rho + w_2$ [136], and the coupling $\pi \int \rho A \frac{dA}{4}$ transforms by

$$\pi \int Sq^2 \left( A \frac{dA}{4} \right) = \pi \int A Sq^2 \left( \frac{dA}{4} \right) = \pi \int A \frac{dA}{4} \frac{dA}{4} , \tag{3.17}$$

which exactly compensates the transformation of $\frac{2\pi}{4} \int A \frac{dA}{4} \frac{dA}{4}$ under flipping the orientation. Thus the entire action can be defined on un-orientable manifolds.

We can also express the Chern-Simons term using the $\mathbb{Z}_4$ valued quadratic form $q_\rho$ as

$$\frac{2\pi}{4} \int A \cup q_\rho \left( \frac{dA}{4} \right) = \frac{2\pi}{4} \int_{\mathrm{PD}(A)} q_\rho \left( \frac{dA}{4} \right) , \tag{3.18}$$

where $\mathrm{PD}(A)$ means the Poincaré dual of $A$. See Ref. [136] for a review of the quadratic form $q_\rho$.

**Bosonic 4+1d SPTs**   $\mathbb{Z}_4$ SPTs have a $\mathbb{Z}_4 \times \mathbb{Z}_4$ classification labeled by $\omega = (k, l)$, that we will define below. One $\mathbb{Z}_4$ factor comes from the group cohomology classification, given

---

[11]These are given by $Sq^n(a \cup b) = \sum_{i+j=n} Sq^i(a) \cup Sq^j(b)$ and $Sq^i Sq^j = \sum_{k=0}^{\lfloor i/2 \rfloor} \binom{j-k-1}{i-2k} Sq^{i+j-k} Sq^k$ respectively. In particular, the latter gives $Sq^1 Sq^1 = 0$.

by the Chern-Simons term described above. In addition, there is also a $\mathbb{Z}_4$ classification of beyond cohomology SPT phases, with action

$$\frac{\pi \ell}{2} \int A p_1(TM) = \frac{2\pi \ell}{4} \int_{\text{PD}(A)} p_1(TM) \, , \tag{3.19}$$

where $\ell = 0, 1, 2, 3 \mod 4$, and $p_1(TM)$ is the first Pontryagin class of the tangent bundle. This means that the domain wall that generates the $\mathbb{Z}_4$ symmetry is decorated with a gravitational theta term.

Note that the possible bosonic invertible topological phases in 4+1d also include the invertible phase with the effective action $\pi \int w_2 \cup w_3$ [134, 137, 138]. However, this phase is independent of the $\mathbb{Z}_4$ symmetry and thus does not affect out discussion. The invertible topological phases with $\mathbb{Z}_4$ symmetry modulo the invertible phases without symmetry have $\mathbb{Z}_4 \times \mathbb{Z}_4$ classification (see *e.g.* [139]), and are labelled by $\omega = (k, \ell)$ as above.

**Fermionic 4+1d SPTs** The 4+1d topological terms for $\mathbb{Z}_4 \times \mathbb{Z}_2^f$ symmetry (where $\mathbb{Z}_2^f$ is the fermion parity symmetry) are classified by $\mathbb{Z}_4$ [140], and the generator can be described by the anomaly of 3+1d massless free Dirac fermion (see *e.g.* [141]) with $\mathbb{Z}_4$ charge 2 and charge 1 (the charges are chosen such that the fermion parity is not identified with the $\mathbb{Z}_2$ subgroup of $\mathbb{Z}_4$ symmetry):

$$I = \frac{2\pi}{4} \int_{5d} \left( \frac{2^3 + 1^3}{3!} A \frac{dA}{4} \frac{dA}{4} \right) - \frac{2\pi}{4} \int_{5d} (2+1) A \frac{p_1}{24} = 3 I_0, \tag{3.20}$$

where

$$I_0 = \frac{2\pi}{8} \int_{5d} A \frac{dA}{4} \frac{dA}{4} - \frac{2\pi}{4} \int_{5d} A \frac{p_1}{24} \, . \tag{3.21}$$

Thus the SPT phases are generated by action $I_0$, which is the $(k, \ell) = (1/2, -1/24)$ term in the previous notation. Note that the action $I_0$ has order four on spin manifolds, matching the $\mathbb{Z}_4$ classification above. This is because on a spin manifold, $\sigma = p_1/3$ is a multiple of 16, where $\sigma$ is the signature of the 4-manifold. Therefore, the second term in (3.21) is order two. Moreover, the $k = 2$ term on spin manifolds is trivial. It follows that we can label the effective actions of the SPT phases by $\omega_f I_0$ with $\omega_f = 0, 1, 2, 3 \mod 4$, where $2 I_0$ is the $\omega = (1, 0)$ term.

### 3.4.2 3+1d SPTs on the domain walls

The above 4+1d $\mathbb{Z}_4$ SPT phases can be described by 3+1d SPT phases that decorate the domain walls of the $\mathbb{Z}_4$ symmetry. Let us characterize these 3+1d SPTs, to determine the anomalies they correspond to at the 2+1d boundary defects.

We begin with the Chern-Simons term, which describes four of the SPTs in the bosonic case and two of the SPTs in the fermionic case. This SPT is given by the action (3.16), with

a coefficient $k = 0, 1, 2, 3 \mod 4$:

$$\text{Generalized FS indicator } k : \quad \frac{2\pi k}{4} \int A \frac{dA}{4} \frac{dA}{4} + \pi k \int A \frac{dA}{4} \rho \,, \qquad (3.22)$$

where $A$ is the background $\mathbb{Z}_4$ gauge field. On the domain wall that generates the symmetry, the $\mathbb{Z}_4$ symmetry becomes $\mathsf{T}^2 = C$, where $C^2 = 1$ is charge conjugation [65]. Again we stress that the bulk theory does not in general have $\mathsf{T}$ symmetry only the defect does. The background gauge fields for these symmetries are related by $A = 2\tilde{B}_1 + \tilde{w}_1$, where $B_1$ is the background for $C$ symmetry and tilde denotes a lift from $\mathbb{Z}_2$ to $\mathbb{Z}_4$. Let us discuss odd and even values of $k$ separately:

- For odd $k$, there is dependence on $\rho$, and we restrict the Wu structure of the bulk to the domain wall. The domain wall is described by the inflow term from $A \to A + d\phi$ for $\phi = 1$:

$$\frac{\pi k}{2} \int q_\rho(y_2) \,, \qquad (3.23)$$

  where $y_2 = dA/4 = d(2\tilde{B}_1 + \tilde{w}_1)/4$ as above. A boundary state is 2+1d $\mathbb{Z}_2$ doubled semion theory, where the boson and semion (or anti-semion) have $C^2 = -1$ and $\mathsf{T}^2 = i$.

- For even $k$, there is no dependence on $\rho$, so the action simplifies from (3.23). The domain wall is described by the inflow term from $A \to A + d\phi$ for $\phi = 1$ as:

$$\frac{\pi k}{2} \int y_2 \cup y_2 \,, \qquad (3.24)$$

  where $y_2 = dA/4 = d(2\tilde{B}_1 + \tilde{w}_1)/4$. For $k = 2$, this is an order two SPT phase for $\mathsf{T}^2 = C$, $C^2 = 1$ symmetry [142]. A boundary state is 2+1d $\mathbb{Z}_2$ toric code where the electric and magnetic particles have $C^2 = -1$ and $\mathsf{T}^2 = i$.

**Bosonic 3+1d SPTs** The above 3+1d $\mathsf{T}^2 = C$ SPTs describe decorated domain walls of the in-cohomology 4+1d $\mathbb{Z}_4$ bosonic SPT phases $\omega = (k, 0)$. The four 4+1d beyond cohomology $\mathbb{Z}_4$ SPT phases $\omega = (0, \ell)$ have domain walls decorated by SPTs corresponding to framing anomalies given by $c_- = 2\ell$ on their 2+1d boundary,[12] due to the gravitational theta term.

**Fermionic 3+1d SPTs** For fermionic theories, the SPT phases are described by 4+1d effective actions labeled by $\omega_f = I_0$ with $\omega_f = 0, 1, 2, 3 \mod 4$. When $\omega_f = 0, 2$, the effective action is the same as the bosonic actions corresponding to $\omega = (0, 0), (1, 0)$, and it represents an anomaly for $\mathsf{T}^2 = (-1)^F C$ symmetry as discussed above. For $\omega_f = 1, 3 \mod 4$, the effective action of the SPT phase contains an $Ap_1$ term, that implies the 3+1d domain wall is decorated with an SPT whose 2+1d boundary has $c_- = \omega_f/4$.

---

[12]The 3+1d term $-\frac{2\pi c_-}{8}\sigma$ for signature $\sigma = p_1/3$ implies that the 2+1d boundary has framing anomaly $c_-$.

## 3.5 Obstructions to symmetric TQFTs

Let us first remark on how $\omega$ and $\omega_f$ modify the classification of symmetric TQFTs, such as those discussed in Section 2.2.1. We consider separately $\omega, \omega_f$ describing SPTs within the group cohomology classification, and those outside of the group cohomology classification.

If $\omega$ describes a 4+1d group cohomology SPT, then it does not present any additional obstruction to a symmetric TQFT. This is because such SPTs always admit a symmetric gapped boundary given by a finite group gauge theory [143] (see also [27] for a review). Thus the gapped boundaries for non-trivial bulk SPT phase can be obtained from the gapped boundaries for the case with trivial bulk SPT by stacking with the finite group gauge theory on the boundary. The boundary non-invertible symmetry then acts as

$$\mathcal{D}'_g = \mathcal{D}_g \otimes \tilde{\mathcal{D}}_g \, , \tag{3.25}$$

where $\tilde{\mathcal{D}}_g$ acts on the finite group gauge theory, while $\mathcal{D}_g$ generates the non-invertible symmetry on the boundary with trivial bulk SPT phase (*i.e.* trivial $\omega, \omega_f$).

### 3.5.1 Obstructions from "beyond group cohomology" SPTs

While the group cohomology SPTs have gapped, symmetric boundaries as described above, there can be obstructions from beyond group cohomology SPTs. We will show that these do not occur for $S$ symmetry, but they do occur for more general $ST^n$ symmetries. For bosonic theories, beyond-cohomology SPTs in 4+1d can be described by a one-dimensional representation of $\mathrm{Aut}(K)$, $\omega_1 \in H^1(\mathrm{Aut}(K), U(1))$, with the 4+1d effective action

$$\int \omega_1 \cup (p_1/3) \, , \tag{3.26}$$

where $p_1$ is the first Pontryagin class of the tangent bundle. We note that since $p_1 = 3\sigma$, and $p_1 = 3\mathcal{P}(w_2) + 2w_1^4 \bmod 4$, when the order of $\omega_1$ is 4 (which applies to $S$ symmetry), the anomaly can be realized by a symmetric TQFT by the inflow construction in [144]. If the order $\omega_1$ does not divide 4, the anomaly may not be realized by a TQFT; if this is the case, it is an example of "symmetry-enforced gaplessness"(see *e.g.* [131, 132] for other examples). Such beyond group cohomology SPTs present obstructions to symmetric gapped boundary.

Let us give another argument using the partition function of 3+1d TQFT. It is known that the partition function of unitary TQFTs without local operators on simply connected spin 4-manifolds is positive [145]. Let us take the 4-manifold to be K3,

$$Z_{\mathrm{TQFT}}(K3) \neq 0 \, . \tag{3.27}$$

On the other hand, since K3 manifold has signature $-16$, $\int_{K3} p_1 = -48$, the anomaly from (3.26) implies that under a transformation $\omega_1 \to \omega_1 + d\alpha$, the partition function transforms by the phase factor

$$Z(K3) \quad \longrightarrow \quad Z(K3) \, e^{-16i\alpha} \, . \tag{3.28}$$

If $16\alpha \notin 2\pi\mathbb{Z}$, the partition function transforms by a non-trivial phase factor under the global symmetry transformation. Thus the partition function vanishes on $K3$ manifold, contracting the positive condition (3.27). We conclude that no symmetric gapped phase can realize the non-invertible symmetry with such "beyond group cohomology" FS indicator.

For instance, if the non-invertible symmetry has the fusion rule of the triality symmetry in [39], then the generalized Frobenius-Schur indicator is given by the beyond group cohomology SPT phase

$$\frac{2\pi}{6} \int A \cup (p_1/3) \, , \tag{3.29}$$

where $A$ is the background field for bulk $\mathbb{Z}_6$ permutation symmetry. This gives an obstruction to realizing the non-invertible symmetry in symmetric gapped phases.

We now proceed to study obstructions to symmetric invertible TQFTs with $S$ symmetry from $\omega, \omega_f$.

## 3.6   Kramers-Wannier duality symmetry in 3+1d

As in the 1+1d example in Section 3.3, we must first determine the 2+1d defect theory that ends the non-invertible duality defect of a 3+1d non-invertible symmetry with 1-form symmetry labeled by $N$ and $p$. We will then check if these theories can cancel the anomalies described above. If so, then the 3+1d non-invertible symmetry is a non-invertible extension of the $\mathbb{Z}_4$ symmetry that trivializes the anomaly. This means that the non-invertible symmetry is anomaly-free.

The non-invertible Kramers-Wannier symmetries in 3+1d that pass the first level obstruction for $S$ gauging are listed in (2.26). These are $N$ that satisfy $N = \alpha^2 + \beta^2$ with integers $\alpha$ and $\beta$ satisfying $\gcd(\alpha, \beta) = 1$. It can be shown that $N$ satisfies this condition if and only if there exists an integer $p$ such that $p^2 = -1$ mod $N$.[13] In fact, each of the Lagrangian subgroups in (2.26) gives a boundary theory described by a $\mathbb{Z}_N$ 1-form symmetry labeled by $N, p$, with action

$$\frac{2\pi p}{2N} \int_X \mathcal{P}(B), \tag{3.30}$$

where $B$ is a background gauge field for the $\mathbb{Z}_N$ 1-form symmetry and

$$\mathcal{P}(B) = \begin{cases} B \cup B - B \cup_1 dB \in H^4(X, \mathbb{Z}_{2N}) & N \text{ even} \\ B \cup B \in H^4(X, \mathbb{Z}_N) & N \text{ odd.} \end{cases} \tag{3.31}$$

These SPTs are invariant under gauging the 1-form symmetry [65]. For a Kramers-Wannier symmetry with 1-form symmetry specified by $N$ and $p$, it was shown in Ref. [65] that the 2+1d defect of the non-invertible duality symmetry $\mathcal{D}$ is described by the minimal

---

[13]One can also show that $N$ is a product of pythagorean primes, which are primes that are 1 mod 4, possibly with a factor of two.

Abelian TQFT $\mathcal{A}^{N,p}$. This is a theory consisting of Abelian anyons $a^q$, $q \in [0, N)$ with $\mathbb{Z}_N$ fusion rules and spins $h[a^q] = \frac{pq^2}{2N}$ mod 1. The $S$ matrix follows from the Abelian fusion and the topological spins, and is given by

$$S_{qq'} = \frac{1}{\sqrt{N}}\exp\left(-\frac{2\pi ip}{N}qq'\right), \tag{3.32}$$

which is unitary if $\gcd(p, N) = 1$ (this is always the case for $p$ satisfying $p^2 = -1$ mod $N$). If $pN$ is even, the theory is bosonic, and if $pN$ is odd, the theory is fermionic. We will consider the cases where $pN$ is even.

Ref. [65] furthermore showed that the duality defect has a $\mathsf{T}^2 = C$ symmetry when $N$ is odd and $p$ is even, and a $\mathsf{T}^2 = C(-1)^F$ symmetry when $N$ is even and $p$ is odd. When $N$ is even, we must add an additional transparent fermion to make the theory time reversal invariant, which is why we considered fermionic SPTs in Section 3.4. Specifically, the time reversal symmetry actions are given by

$$\text{Even } N : a^q \to a^{p^{-1}q} = a^{-pq} \qquad \text{Odd } N : a^q \to a^{-pq}f^q, \tag{3.33}$$

where $f$ is the transparent fermion. It is straightforward to check that both symmetries take $h[a^q] \to -h[a^q]$, and that applying them twice takes $a^q \to a^{-q}$. We will check whether the $\mathsf{T}^2 = C$ symmetry in $\mathcal{A}^{N,p}$ for odd $N$ and the $\mathsf{T}^2 = C(-1)^F$ symmetry in $\mathcal{A}^{N,p}$ for even $N$ are anomalous, to cancel the anomalies of Section 3.4.2.

### 3.6.1 Anomalies in $\mathcal{A}^{N,p}$ with odd $N$

Let us begin with theories with odd $N$. Ref. [146] showed that the $\mathsf{T}^2 = C$ symmetry in $\mathcal{A}^{N,p}$ for $N = m^2 + n^2$ with even $p = m$ and odd $n$ (and $\gcd(m, n) = 1$) is anomaly-free, by explicitly writing down a symmetric gapped phase under $\mathbb{Z}_4^{\mathsf{T}}$. The same method can be used to prove that $\mathcal{A}^{N,p}$ with odd prime $N$ are all anomaly-free.[14] For more general $N, p$ we will show that the $\mathsf{T}^2 = C$ symmetry is anomaly-free by studying the structure of the time-reversal symmetry after gauging $C$. We will show that the resulting theory has a $\mathsf{T}^2 = 1$ symmetry. Because the gauging does not cause the time-reversal symmetry to be extended (to form a 2-group) or become non-invertible, we conclude that the original $\mathsf{T}^2 = C$ symmetry is not anomalous [27].

---

[14]The basic idea is to use the $U(1)$ Chern-Simons description of the anyon theory, with $K$ matrix $\left(\begin{smallmatrix} m & n \\ n & -m \end{smallmatrix}\right)$, and find null vectors for the $K$ matrix describing the interface edge theory between two regions related by reflection. This method works for any $K$ matrix of the form $\left(\begin{smallmatrix} m & n \\ n & -m \end{smallmatrix}\right)$, but in general, this describes $\mathcal{A}^{N,m}$ and $m \neq p$. However, if $N$ is prime, one can show that there always exists an integer $x$ such that $m = px^2$ mod $N$, so this $K$ matrix also describes $\mathcal{A}^{N,p}$. In other words, $\mathcal{A}^{N,m}$ is equivalent to $\mathcal{A}^{N,p}$ upon a relabeling of the anyons. To show that such $x$ exists, note that $m = px^2$ mod $N$ means that $-pm$ (and therefore $pm$) is a quadratic residue of $N$. From $m^2 + n^2 = N$ we get $n = pm$ mod $N$, where $n$ is odd. $n$ is always a quadratic residue of $N$ by quadratic reciprocity.

**Gauging the $C$ symmetry**   Let us gauge the charge conjugation symmetry without stacking any additional invertible phases. The anyons in the gauged theories can be obtained using the methods in *e.g.* [110,147], and they are as follows:

- $a^0 = \mathbf{1}$ and the charge conjugation defect $\chi$ are invariant under the symmetry. They split into $\mathbf{1}, \epsilon$ and $\chi_+, \chi_-$ respectively. $\mathbf{1}$ and $\epsilon$ have quantum dimension 1 and spin 0, and $\chi_+$ and $\chi_-$ have quantum dimension $\sqrt{N}$ and spin $\frac{c_-}{16}$ and $\frac{c_-}{16} + \frac{1}{2}$, where $c_-$ is the framing anomaly. They obey the fusion rule $\chi_+ = \epsilon \chi_-$. In the particular case of $\mathcal{A}^{N,p}$ with odd $N$ and $p^2 = -1$ mod $N$, $c_- = 0$ [65], so the spins of $\chi_+$ and $\chi_-$ are 0 and $\frac{1}{2}$ respectively. Note that $\epsilon$ is represented by the Wilson line for the $C$ symmetry $e^{i \int a'}$, where $a'$ is the background gauge field for $C$.

- The other anyons in $\mathcal{A}^{N,p}$ form orbits of size two under the symmetry, resulting in anyons of the gauged theory with quantum dimension 2 and spin $\frac{pq^2}{2N}$. We label these anyons by $a + a^{N-1}, a^2 + a^{N-2}, \cdots a^{\frac{N-1}{2}} + a^{\frac{N+1}{2}}$.

One can verify that the total quantum dimension is $D^2 = \sum_a d_a^2 = 2 + 2N + \frac{N-1}{2} \cdot 4 = 4N$.

**Gauging the time-reversal symmetry: anomaly indicators**   We can check whether the time-reversal forms a 2-group with the $\mathbb{Z}_2$ 1-form symmetry generated by $\epsilon$ by computing the anomaly indicators for time-reversal symmetry [111,148–150]. These indicators detect obstructions to gauging the time-reversal symmetry. There are two anomaly indicators:

$$\eta_1 = \frac{1}{D} \sum_{a \in \mathcal{C}} d_a^2 e^{2\pi i h[a]} = e^{\frac{2\pi i c_-}{8}} \qquad \eta_2 = \frac{1}{D} \sum_{a \in \mathcal{C}_\mathsf{T}} d_a \mathsf{T}_a^2 e^{2\pi i h[a]}, \tag{3.34}$$

where $\mathcal{C}$ is the set of all anyons in the theory and $d_a$ is the quantum dimension of anyon $a$. In the definition of $\eta_2$, we use $\mathcal{C}_\mathsf{T} \subset \mathcal{C}$ to denote anyons that are not permuted under the time-reversal symmetry, and $\mathsf{T}_a^2 = \pm 1$ indicates whether $a$ is Kramers singlet or doublet.

These two anomaly indicators take value in $\pm 1$ when the time-reversal symmetry does not participate in a 2-group with $\epsilon$ [111]. $\eta_1 = 1$ because $c_- = 0$ mod 8 [65], so all we need to do is compute $\eta_2$. The only anyons in $\mathcal{C}_\mathsf{T}$ are $1, \epsilon, \chi_+$, and $\chi_-$, because these are the only anyons with spin 0 or $\frac{1}{2}$ mod 1. Furthermore, $\mathsf{T}_\epsilon^2 = -1$ so $\mathsf{T}_{\chi_+}^2 = -\mathsf{T}_{\chi_-}^2$ from the fusion rule $\chi_+ = \epsilon \chi_-$. Putting this together with the spins listed above, we find

$$\eta_2 = \frac{1}{2\sqrt{N}} \left( 1 - 1 + \sqrt{N} \mathsf{T}_{\chi_+}^2 - (-1)\sqrt{N} \mathsf{T}_{\chi_+}^2 \right) = \mathsf{T}_{\chi_+}^2 = \pm 1. \tag{3.35}$$

Therefore, the time-reversal symmetry does not participate in a 2-group with the 1-form symmetry generated by $\epsilon$. The value of $\eta_2$ depends on the fractionalization classes for the one-form symmetry generated by $\epsilon$, classified by $H^2(\mathbb{Z}_2, \mathbb{Z}_2) = \mathbb{Z}_2$. The fractionalization class determines $\mathsf{T}_{\chi_+}^2$ because $\chi_\pm$ have $\pi$ mutual statistics with $\epsilon$. We see that there is always a fractionalization that allows us to gauge the entire $\mathsf{T}^2 = C$ symmetry. Changing the

fractionalization amounts to adding the local counterterm $B_1 \cup w_1^2$, where $B_1$ is the gauge field for $C$ symmetry. Therefore, the anomaly can be cancelled by local counterterm, and we conclude that the $\mathsf{T}^2 = C$ symmetry is non-anomalous in $\mathcal{A}^{N,p}$ theories with odd $N$ and $p^2 = -1 \bmod N$. Moreover, since $c = 0 \bmod 8$, the theory does not have framing anomaly.

Because the theory has neither a $\mathsf{T}^2 = C$ anomaly nor a framing anomaly, it cannot cancel any of the anomalies of $\omega$ discussed in Section 3.4.2. Therefore, the non-invertible symmetry with odd $N$ and even $p$ is anomalous when the generalized FS indicator $\omega$ is nontrivial.

**Theorem 1** *The 3+1d Kramers-Wannier (S gauging) non-invertible symmetry with $\mathbb{Z}_N$ 1-form symmetry where $N$ is odd is anomalous if and only if the generalized FS indicator $\omega \in \mathbb{Z}_4 \times \mathbb{Z}_4$ is nontrivial.*

### 3.6.2   Anomalies in $\mathcal{A}^{N,p}$ with even $N$

$\mathcal{A}^{N,p}$ with even $N$ does not have a $\mathsf{T}^2 = C$ symmetry. We will show that the time-reversal symmetry in $\mathcal{A}^{N,p}$ with even $N$ either participates in a 2-group with the $\mathbb{Z}_2$ subgroup of the $\mathbb{Z}_N$ 1-form symmetry, or we can add a transparent fermion by tensoring the theory with $\{1, f\}$ to make the symmetry $\mathsf{T}^2 = (-1)^F C$. We will consider the latter, and determine whether or not the $\mathsf{T}^2 = (-1)^F C$ symmetry is anomalous in a way that cancel the anomalies of Section 3.4.2.

**Time-reversal symmetry and 2-group**   The time-reversal symmetry in the $\mathcal{A}^{N,p}$ theory is given by the permutation

$$q \to pq, \quad p^2 = -1 \bmod N . \tag{3.36}$$

Under the time-reversal transformation, the spin changes into

$$\frac{pp^2 q^2}{2N} = \frac{p(-1+N)q^2}{2N} = -\frac{pq^2}{2N} + \frac{q}{2} , \tag{3.37}$$

where we used $p^2 = -1 + N \bmod 2N$ for even $N$ (note in particular that there does not exist $p$ satisfying $p^2 = -1 \bmod 2N$ for even $N$ because $2N$ is divisible by four). Therefore, the theory has time-reversal symmetry $\mathsf{T}$ obeying $\mathsf{T}^2 = C$ that simultaneously shifts the spin of the odd charge $q$ by a half.

This kind of time-reversal symmetry is discussed in [136], and it participates in a 2-group with $\mathbb{Z}_2$ subgroup of the $\mathbb{Z}_N$ 1-form symmetry, with the Postnikov class being the third Wu class $w_1 w_2$. For an introduction to higher group symmetry, see *e.g.* [111, 151].

Alternatively, we can add transparent fermion by tensoring the theory with $\{1, f\}$, where the fermion $f$ satisfies $\mathsf{T}^2 = (-1)^F$. Then $w_2$ is trivialized, and the 2-group becomes the tensor product of the 1-form symmetry and the time-reversal symmetry that satisfies $\mathsf{T}^2 = (-1)^F C$, $C^2 = 1$. This means that the permutation action of the time-reversal symmetry on the anyons

must mix the transparent fermion with the anyons in the $\mathcal{A}^{N,p}$ theory [152]. The relevant spacetime structure is called "Epin" in [142], where $w_1^2$ and $w_2$ are both exact.

**Gauging the $C$ symmetry: time-reversal symmetry becomes non-invertible**  As in the odd $N$ cause, we will probe the anomaly of $\mathsf{T}^2 = (-1)^F C$ by gauging the $C$ symmetry. Since the transparent fermion does not participate in the $C$ symmetry, we can focus on gauging the $C$ symmetry in the bosonic $\mathcal{A}^{N,p}$ theory. Denote the framing anomaly of the bosonic $\mathcal{A}^{N,p}$ theory by $c(N,p)$, which is an odd integer. Gauging the $C$ symmetry without additional invertible phase, using the methods in *e.g.* [110, 147], gives a theory with the following anyons:

- $a^0 = \mathbf{1}$ and the $C$ defect $\chi$ are invariant under the symmetry. As in the odd $N$ case, they split into $\mathbf{1}, \epsilon$ (with quantum dimension 1 and spin 0) and $\chi_+, \chi_-$ (with quantum dimension $\sqrt{N/2}$ and spin $\frac{c(N,p)}{16}$, $\frac{c(N,p)}{16} + \frac{1}{2}$) respectively. $\chi_\pm$ obey the fusion rule $\chi_+ = \epsilon\chi_-$. Again, $\epsilon$ is represented by the Wilson line for the $C$ symmetry.

- $a^{N/2}$ is also invariant under the symmetry and leads to another $C$ defect $\xi$. These split into $s_+, s_-$ and $\xi_+, \xi_-$ respectively. $s_\pm$ have quantum dimension 1 and spin $\frac{pN}{8}$ mod 1, and $\xi_\pm$ have quantum dimension $\sqrt{N/2}$ and spin of spin $\frac{c(N,p)}{16}$ ($\xi_+$), $\frac{c(N,p)}{16} + \frac{1}{2}$ ($\xi_-$). They have the fusion rules $s_+ = \epsilon s_-$ and $\xi_+ = \epsilon\xi_-$.

- The other anyons in $\mathcal{A}^{N,p}$ form orbits of size two under the symmetry, resulting in anyons in the gauged theory with quantum dimension 2 and spin $\frac{pq^2}{2N}$ mod 1. We label these anyons by $a + a^{N-1}, a^2 + a^{N-2}, \cdots a^{N/2-1} + a^{N/2+1}$.

The total quantum dimension is $D^2 = 4 + 4 \cdot (N/2) + (N/2 - 1) \cdot 2^2 = 4N$, as expected. For instance, when $N = 2$, $\mathcal{A}^{2,1} = U(1)_2$, and we only have the first two kinds of anyons. The gauged theory has 8 anyons, all with quantum dimension 1, and spins $0, 0, 1/4, 1/4, 1/16, 1/16, 9/16, 9/16$. This is precisely $U(1)_8$, in agreement with [147, 153].

Notice that the gauged theory does not have an invertible time-reversal symmetry. Specifically, $c(N,p)$ is odd for odd $N$, so there are no anyons with spins opposite of those of $\chi_\pm, \xi_\pm$. We will see that there is instead a non-invertible time-reversal symmetry. Because the time-reversal symmetry becomes non-invertible after gauging $C$, we conclude that the $\mathsf{T}^2 = C$ (or $\mathsf{T}^2 = C(-1)^F$ if we add the transparent fermion) must be anomalous [27].

Let us show that the time-reversal symmetry becomes non-invertible after gauging the $C$ symmetry. Denoting the gauged theory by $O^{N,p}$, we will show that the theory is invariant under the following time-reversal symmetry $\mathsf{T}'$:

$$\text{non-invertible time-reversal}: \mathsf{T}' = \mathsf{T} \circ S_p[X], \qquad (3.38)$$

where

$$S_p[X] = \frac{X \times (\mathbb{Z}_2)_{2c(N,p)}}{\mathbb{Z}_2} = \begin{cases} \frac{X \times (\mathbb{Z}_2)_2}{\mathbb{Z}_2} & p = 1 \bmod 4 \\ \frac{X \times (\mathbb{Z}_2)_{-2}}{\mathbb{Z}_2} & p = 3 \bmod 4 \end{cases}, \qquad (3.39)$$

Non-invertible time reversal symmetry

$$O^{N,p} \qquad\qquad T \circ S[O^{N,p}] = O^{N,p}$$

Figure 2: Non-invertible time reversal symmetry of $O^{N,p}$, which is the $\mathcal{A}^{N,p}$ after gauging $C$. $\mathsf{T}$ means reversing the orientation, and $S$ means coupling to a $\mathbb{Z}_2$ gauge theory: $S[X] := \frac{X \times (\mathbb{Z}_2)_{2p}}{\mathbb{Z}_2}$ for general 2+1d theories $X$ that has non-anomalous $\mathbb{Z}_2$ one-form symmetry. We have omitted the transparent fermion $\{1, f\}$ in the figure.

where $(\mathbb{Z}_2)_{2c(N,p)}$ is the fermionic Abelian $\mathbb{Z}_2$ gauge theory with Chern-Simons level $c(N,p)$ [153], whose magnetic charge has spin $-c(N,p)/8$ mod 1. The diagonal $\mathbb{Z}_2$ quotient is generated by the tensor product of a non-anomalous 1-form symmetry in $X$ and the $\mathbb{Z}_2$ electric charge, which are both bosons. In (3.39), we used the property that the chiral central charge satisfies $c(N,p) = p$ mod 4 for $N = 2$ mod 4 [124], which simplifies $(\mathbb{Z}_2)_{2c(N,p)} = (\mathbb{Z}_2)_{2p}$ by the mod 8 periodicity of the $\mathbb{Z}_2$ topological term (see *e.g.* [153]).

For $X = O^{N,p}$, we choose the non-anomalous 1-form symmetry to be $\epsilon$, so the $\mathbb{Z}_2$ quotient is generated by the tensor product of $\epsilon$ and the $\mathbb{Z}_2$ electric charge of $(\mathbb{Z}_2)_{2p}$. The theory is invariant under (3.39) because $S_p[O^{N,p}]$ flips the spins of $\chi_\pm$ and $\xi_\pm$, and $\mathsf{T}$ flips the spins of all the anyons, in particular flipping back the spins of $\chi_\pm$ and $\xi_\pm$, bringing the theory back to $O^{N,p}$. This time-reversal symmetry is non-invertible: the domain wall that generates the time-reversal symmetry is decorated with the topological boundary condition of the Abelian $\mathbb{Z}_2$ Chern-Simons theory $(\mathbb{Z}_2)_{2p}$ (see Figure 2). The fusion rule can be computed using *e.g.* [26, 28, 29, 35, 39, 49, 60], and is as follows:

$$\mathsf{T}' \times \overline{\mathsf{T}'} = \frac{1}{\mathcal{M}} \left( 1 + i^{p \int q_\rho(B_1)} \right) , \tag{3.40}$$

where $q_\rho(B_1)$ is the $\mathbb{Z}_4$ valued quadratic form for the $\mathbb{Z}_2$ gauge field $B_1$ for the $C$ symmetry, and $\mathcal{M}$ is an overall normalization numerical factor. The operator $i^{p \int q_\rho(B_1)}$ represents $p$ copies (note that $p$ is odd) of the Kitaev chain [125, 126].

We remark that in an upcoming work [154], we will investigate other 2+1d TQFTs and Chern-Simons matter theories with similar non-invertible time-reversal symmetry. Examples of non-invertible time-reversal symmetry in 3+1d are discussed in *e.g.* [44].

**Diagnosis for the anomaly in $\mathcal{A}^{N,p}$** Because the time-reversal becomes non-invertible after gauging $C$, we conclude that the original $\mathsf{T}^2 = (-1)^F C$ in $\{1, f\} \times \mathcal{A}^{N,p}$ is anomalous. We now determine which anomalies it can cancel out of those described in Section 3.4.2.

The fact that the non-invertible time-reversal symmetry requires coupling to $(\mathbb{Z}_2)_{2c(N,p)}$, which has the action $\frac{\pi c(N,p)}{4} \int B_1 dB_1$ (the factor $1/4$ can be defined in fermionic theories), implies that the time-reversal symmetry in $\mathcal{A}^{N,p}$ has mixed anomaly with the charge conjugation symmetry, described by the anomaly term

$$\mathcal{A}_{\text{bulk}} \supset -\frac{\pi c(N,p)}{8} \int dB_1 dB_1 = -\frac{\pi c(N,p)}{2} \int_{4d} \left(\frac{2dB_1}{4}\right)^2 . \tag{3.41}$$

To see this, we note that reversing the orientation produces the term $\frac{\pi c(N,p)}{4} \int_{4d} dB_1 dB_1 = \frac{\pi c(N,p)}{4} \int_{3d} B_1 dB_1$. Therefore, after gauging the charge conjugation symmetry, the time-reversal symmetry becomes the non-invertible time-reversal symmetry $\mathsf{T}' = \mathsf{T} \circ S$. It follows that for even $N$, the theory $\mathcal{A}^{N,p}$ has time-reversal anomaly (for the trivial fractionalization class) is described by the bulk term

$$-\frac{\pi c(N,p)}{2} \int q_\rho \left(\frac{dA}{4}\right) = -\frac{\pi p}{2} \int q_\rho \left(\frac{dA}{4}\right) = \pm \frac{\pi}{2} \int q_\rho \left(\frac{dA}{4}\right) \text{ mod } 2\pi , \tag{3.42}$$

where $A = 2\tilde{B}_1 + \tilde{w}_1$, and we used $c(N,p) = p$ mod 4. By reversing the orientation (or changing the Wu$_3$ structure $\rho$), the coefficients can be $\pm$. The above anomaly corresponds to $\omega = (1,0), (3,0)$ or $\omega_f = 2$ mod 4.

**Anomalies from changing the fractionalization class** Does the anomaly of $\mathcal{A}^{N,p}$ indicate that the 3+1d Kramers-Wannier symmetry is only anomaly free if $\omega_f = 2$ mod 4? This is not the case, because we can change the fractionalization class. Here, we will show that fractionalization of the $\mathsf{T}^2 = (-1)^F C$ symmetry can cancel the above anomaly, allowing the 3+1d Kramers-Wannier symmetry symmetry with 1-form symmetry $\mathbb{Z}_N$ with even $N$ to be anomaly-free, even with trivial $\omega_f$.

Since $N$ is even, the theory has $\mathbb{Z}_2$ subgroup of the 1-form symmetry generated by $a^{N/2}$, which is invariant under charge conjugation. There can be non-trivial fractionalization class for the $\mathsf{T}^2 = (-1)^F C$ symmetry on this 1-form symmetry. Here, we will regard this as a $R^2 = C$ symmetry in the Euclidean signature spacetime. We can describe the fractionalization using background fields.

Denote the background field for $C$ and $R$ by $B_1$ and $w_1$, and the background for the $\mathbb{Z}_2$ subgroup of the 1-form symmetry by $B_2$. Then we can consider the fractionalization class from activating the background [111, 113]

$$B_2 = \frac{dA}{4} \text{ mod } 2 = \text{Bock}(A) , \tag{3.43}$$

where $A := 2\tilde{B}_1 + \tilde{w}_1$ is a $\mathbb{Z}_4$ 1-cocycle, tilde denotes lift from $\mathbb{Z}_2$ to $\mathbb{Z}_4$, and it is a $\mathbb{Z}_4$ cocycle $dA = 0$ mod 4 since $dB_1 = d\tilde{w}_1/2 = \text{Bock}(w_1)$ mod 2 as describing $R^2 = C$ symmetry extension of $R$ by $C$. The operation Bock is the Bockstein homomorphism, see $e.g.$ [111] for

a review. Since the $\mathbb{Z}_2$ subgroup of the 1-form symmetry is generated by an anyon of spin $\frac{p(N/2)^2}{2N} = \frac{N}{2}\frac{p}{4} = p/4 \mod 1 = \pm 1/4 \mod 1,$[15] the 1-form symmetry is anomalous. The above fractionalization class induces an anomaly for the symmetry given by [113]

$$\pm\frac{2\pi}{4}\int q_\rho(B_2) = \pm\frac{2\pi}{4}\int q_\rho\left(\frac{dA}{4}\right) , \tag{3.44}$$

where the sign is positive if $(N/2)p = 1 \mod 4$ and negative if $(N/2)p = 3 \mod 4$. $q$ is the quadratic function for the Wu$_3$ structure $\rho$ that satisfies $d\rho = w_1 w_2$ inhered from the trivialization of $w_2$, as discussed in Section 3.4.1 (for a review, see *e.g.* [136]). We can flip the sign by fractionalizing on $a^{N/2}f$ rather than $a^{N/2}$. Therefore the fractionalizations realize the $\omega = (1,0),(3,0)$ or $\omega_f = 2 \mod 4$ anomalies. This means that changing the fractionalization class can trivialize the anomaly, so the 3+1d Kramers-Wannier non-invertible symmetry with 1-form symmetry with even $N$ does not need $\omega_f = 2 \mod 4$; it is also anomaly-free with $\omega_f = 0 \mod 4$.

$\omega_f = \pm 1$ **mod 4 anomalies** According to the discussion above, $\mathcal{A}^{N,p}$ with even $N$ and odd $p$ can can produce the $\mathsf{T}^2 = (-1)^F C$ anomalies labeled by $\omega_f = 0, 2 \mod 4$. On the other hand, since the tensor product of the theory $\mathcal{A}^{N,p}$ and the transparent fermion has framing anomaly quantized in units of $1/2$ [116], it cannot realize the anomaly of $\omega_4 = 1, 3 \mod 4$, which requires framing anomaly $\pm 1/4$. In summary,

**Theorem 2** *The 3+1d Kramers-Wannier (S gauging) non-invertible symmetry with $\mathbb{Z}_N$ 1-form symmetry where $N$ is even is anomalous if and only if the generalized (fermionic) FS indicator $\omega_f \in \mathbb{Z}_4$ is odd. Otherwise, it is anomaly-free.*

# 4 Lattice models with non-invertible symmetry $ST^n$

In this section, we will give a method for constructing lattice models invariant under $S$ gauging of the $\mathbb{Z}_N$ 1-form symmetry. These models are akin to the Ising model (and more general clock models) at criticality. We will also discuss $ST^n$ gauging, which is gauging with an additional topological term $n$ for the $\mathbb{Z}_N$ two-form gauge field [124] (see Eq. 2.12).

We will give an example of a model with $ST^4$ symmetry, whose defects have fusion rules different but related to those written in (2.13).

---

[15]Since $\gcd(N,p) = 1$, $p$ is odd for even $N$, and $p^2 = -1 \mod N$ implies that $N = 2 \mod 4$. Moreover, one can show that $N = 2 \mod 8$ because $N$ is has a single factor of 2, and all its other prime factors are Pythagorean primes, of the form $4k + 1$ where $k$ is an integer [28].

## 4.1 Lattice models with $S$ symmetry

We consider lattice models in $3 + 1$ spacetime dimensions, where space is put on a cubic lattice $[0, 1]^3$. We assign to each edge of the lattice a local "matter" Hilbert space, where the symmetry acts, and we assign to the faces Hilbert spaces associated with the gauge fields. For the theory to be invariant under gauging the 1-form symmetry, the faces must be dualized to be edges on the dual lattice. This is the case here because the dual of a cubic lattice is also a cubic lattice, and the faces get mapped to the edges of the dual lattice and vice versa.[16]

We will focus on lattice models with $\mathbb{Z}_N$ 1-form symmetry, but our methods can easily be generalized to other finite, Abelian groups $\prod_i \mathbb{Z}_{N_i}$. The operators acting on the edges are generated by the $\mathbb{Z}_N$ generalizations of the Pauli operators, $X_e$ and $Z_e$, which obey

$$Z_e^N = 1 \qquad X_e^N = 1 \qquad X_e Z_e = Z_e X_e e^{2\pi i/N}. \tag{4.1}$$

The operators acting on the faces are generated by $\tilde{X}_f$ and $\tilde{Z}_f$, which obey the same relations (4.1). The 1-form symmetry is generated by $\prod_e X_e^{\sigma_e}$ on closed surfaces, where $\sigma_e = \pm 1$ depending on the orientation of the edge $e$ (see the vertex term in Figure 3). A model invariant under $S$ gauging the symmetry can be constructed as follows.

We start with the model with Hamiltonian $H_0$ built out of $X_e, Z_e$, which has symmetry as described above acting on the edges of the cubic lattice. Gauging the symmetry as described in Section 4.2 produces a Hamiltonian $\tilde{H}_1$, in terms of operators $\tilde{X}_f, \tilde{Z}_f$. In the gauged model, the symmetry is generated by the Wilson operators given by $\prod_f \tilde{Z}_f^{\sigma_f}$ on closed surfaces, where again $\sigma_f = \pm 1$ depending on the orientation of the face $f$ (see the cube term in Fig. 5). For the model to be invariant under gauging, we need to identify the generator with the original symmetry generator $\prod_e X_e^{\sigma_e}$. To do so, we shift the operators on the faces to the edges. In particular we map the symmetry generator $\prod_f \tilde{Z}_f^{\sigma_f}$ to $\prod_e X_{s_k}^{\sigma_e}$. In general, this involves a "half translation, like in the 1+1d Ising chain in [22].[17] We leave the complete study of the fusion rule for the defects on the lattice to future work.

Denoting the resulting Hamiltonian by $H_1$, we then consider the interpolation between the two Hamiltonians given by

$$H = J_0 H_0 + J_1 H_1 \tag{4.2}$$

When $J_0 \gg J_1$, the model $H$ describes the original theory $H_0$, while for $J_1 \gg J_0$, $H$ describes the $S$ gauged theory. At $J_0 = J_1$, we expect the theory to be self-dual with $X \leftrightarrow Z$.

---

[16]This discussion can be generalized to $d + 1$ spacetime dimensions with a $k$-form symmetry, where the matter degrees of freedom reside on the $k$-simplices and the gauge fields on the $(k + 1)$-simplices. The requirement for gauging the $k$-form symmetry to produce a dual $k$-form symmetry is then $(d - k) = k + 1$ so the spacetime dimension must be even, reproducing the result from field theory.

[17]We thank Shu-Heng Shao for bringing up this point.

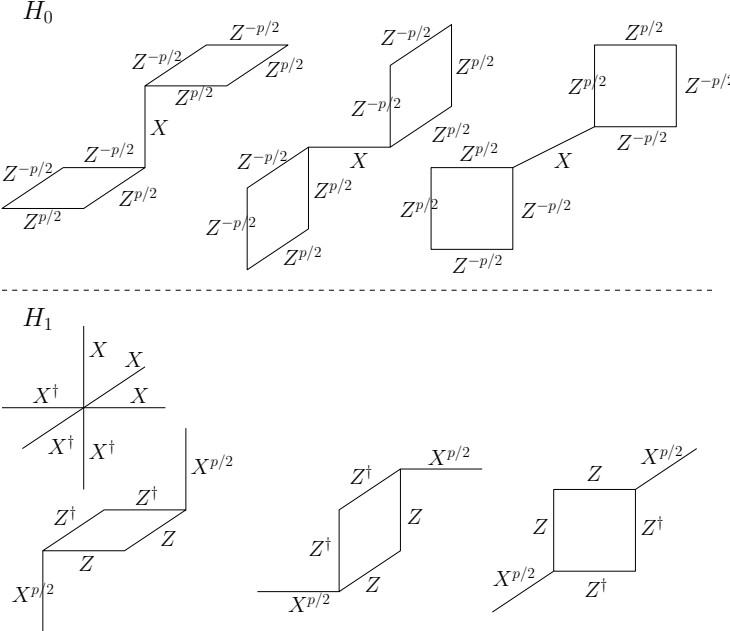

Figure 3: After gauging the 1-form symmetry generated by $\prod_e X_e^{\sigma_e}$ surfaces (vertex term of $H_1$), and then shifting from the dual lattice (faces) to the original lattice (edges) with $X \leftrightarrow Z$, $H_0$ gets mapped to $H_1$. $H_0$ describes the SPT of class $p$ with $\mathbb{Z}_N$ 1-form symmetry [155] while $H_1$ is dual to the 2-form gauge theory with topological action $p$ in Figure 5 on the dual lattice. Here, we show the lattice models for even $p$ for simplicity, the model for odd $p$ can be similarly constructed and is given in [155]. While $H_0$ and $H_1$ are individually commuting Hamiltonians, $H = J_0 H_0 + J_1 H_1$ is not commuting for nonzero $J_0$ and $J_1$.

## 4.2 Microscopic model

A lattice model with a 1-form symmetry generated by closed surfaces $\prod_e X_e^{\sigma_e}$ is given by the bottom row of Figure 3 [155].[18] Note that Fig. 3 shows lattice models for even classes (even $p$) of the 1-form SPTs. When $N$ is odd, these are the most general bosonic 1-form SPTs. When $N$ is even, there are also bosonic SPTs labeled by odd classes. Ref. [155] discusses lattice models for these SPTs. They are relatively complex so we will focus on even classes for simplicity. Let us call the model for the SPT in Fig. 3 $H_0$. $S$ gauging the 1-form symmetry consists of five steps.

1. We introduce $N$-dimensional gauge degrees of freedom on the faces, with local operators generated by $\tilde{X}_f, \tilde{Z}_f$.

2. If the Hamiltonian terms do not commute with the Gauss laws in Figure 4 with $n = 0$ (we will discuss $n \neq 0$ in Section 4.4), which happens when $p \neq 0$, we need to apply

---

[18]While Ref. [155] specified $N$ to be even, the model also works for $N$ odd. Here we allow $N$ to be odd, since we take the class label to be even integer $p$.

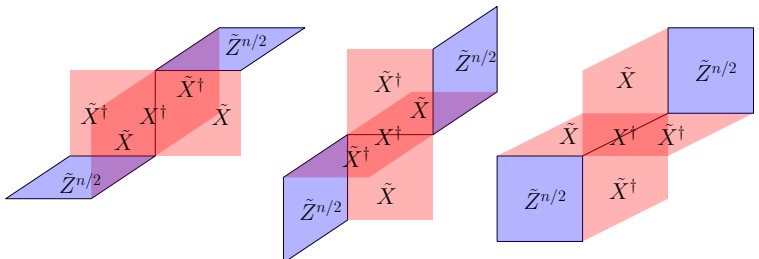

Figure 4: Modified Gauss law constraint that implements $ST^n$ gauging. Each of these terms are set to be 1, so the symmetry transformation on a matter field gets mapped to transformations on neighboring gauge fields. Here we list the model for even $n$, while the model for odd $n$ can be obtained similarly from the SPT models in [155].

   minimal coupling. This modifies the Hamiltonian terms by $\tilde{Z}_f^p$ on each of the two faces to make them commute with the Gauss law terms.

3. We implement the Gauss laws in Figure 4 with $n = 0$ to replace matter operators with gauge field operators.

4. We "integrate out the matter" by fixing the gauge $Z_e = 1$ for all edges. The resulting Hamiltonian is $\tilde{H}_1$

5. We implement zero gauge flux by adding the closed surface in Figure 5 to the Hamiltonian. This generates the dual 1-form symmetry.

We then shift $\tilde{H}_1$ to $H_1$, so its operators act on the edges of the original lattice, and use $X \leftrightarrow Z$. The resulting Hamiltonian is given by the latter two rows of Figure 3. Similarly, gauging $H_1$ via the above five steps gives $\tilde{H}_0$, which after shifting back to the original lattice and using $X \leftrightarrow Z$ gives $H_0$. Note that $H_0$ and $H_1$ are individually commuting, but $H = J_0 H_0 + J_1 H_1$ is not commuting. If we were to shift $H_0$ onto the dual lattice rather than $\tilde{H}_1$ onto the original lattice, then we get a 2-form gauge theory in a transverse field, where the transverse field is given by the 1-form SPT:

$$H = J_0 \tilde{H}_0 + J_1 \tilde{H}_1 . \tag{4.3}$$

where $\tilde{H}_1$ is given by Figure 5. The $S$ gauging of the $p = 0$ case, where $H_0$ is a trivial paramagnet and $\tilde{H}$ is a trivial transverse field, was studied in Appendix B of [28].

## 4.3 Dynamics

As briefly mentioned in Section 4.1, we can infer some regimes of $H$:

- At $J_0 \gg J_1$, we can ignore the term $H_1$, and the theory describes the class $p$ SPT phase with $\mathbb{Z}_N$ one-form symmetry.

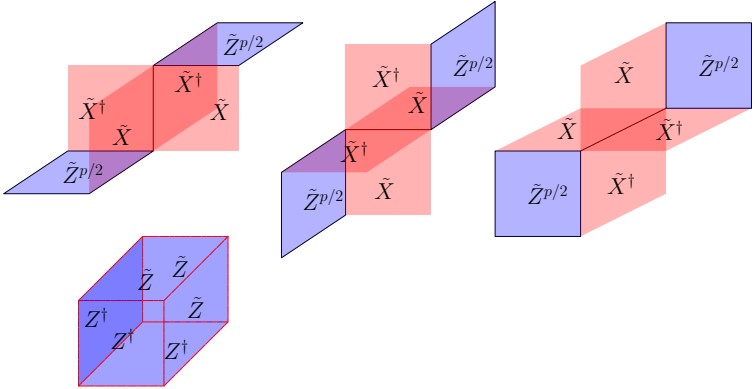

Figure 5: Lattice Hamiltonian for pure 2-form gauge theory with topological action $p$ on cubic lattice. Each face has a $\mathbb{Z}_N$ degree of freedom, acted by the generalization of Pauli operators that satisfying (4.1). The model can be obtained by gauging the 1-form symmetry in $\mathbb{Z}_N$ 1-form symmetry SPT phase of class $p$. The first row is the Gauss law term, while the second row is the flux term. Here we list the model for even $p$, while the model for odd $p$ can be obtained similarly from the SPT models in [155].

- At $J_1 \gg J_0$, we can ignore the term $H_0$, and the theory describes twisted $\mathbb{Z}_N$ 2-form lattice gauge theory with topological action $n$. For $\gcd(N, n) \neq 1$ there are deconfined excitations and non-trivial topological order. For $\gcd(N, p) = 1$, the theory describes a $\mathbb{Z}_N$ 1-form SPT phase of class $-p^{-1}$, but with a surface topological order $\mathcal{A}^{N,p}$, which can have nontrivial chiral central charge $c$ [2, 156]. When the boundary theory has $c \neq 0 \bmod 8$, it can only be obtained from the trivial Hamiltonian $H_0$ with $p = 0$ by a nontrivial quantum cellular automaton (QCA) [157].

We see that even for $p^2 = -1 \bmod N$, where the phases are the same for $J_0 \gg J_1$ and $J_1 \gg J_0$, the two models may differ because their boundaries have different $c$ (mod 8). When $N$ is odd, $\mathcal{A}^{N,p}$ has $c = 0 \bmod 8$ if $p^2 = -1 \bmod N$ [65], but when $N$ is even, $c \neq 0 \bmod 8$.
[19] On the lattice, $H_0$ and $H_1$ differ by QCA even though they describe the same phase.

$H_0$ and $H_1$ are individually commuting Hamiltonians, but $H$ is not commuting for nonzero $J_0$ and and $J_1$. When $\gcd(n, N) \neq 1$, we expect a confinement-deconfinement transition at intermediate values of $J_0$ and $J_1$. For $\gcd(n, N) = 1$, we still expect a quantum phase transition between different SPT phases, except in the case $p^2 = -1 \bmod N$. For $N$ even and $p^2 = -1 \bmod N$, there would at least be a surface transition due to the change in $c$. On the other hand for $N$ odd, it does not seem necessary for the system to undergo a phase transition.

When $p = 0$, the model reduces to the toric code in a transverse field studied in [28],

---

[19]Previously, we considered fermionic systems when $N$ is even, but here we restrict to bosonic lattice models for simplicity.

where it is shown that the model is invariant under gauging $\mathbb{Z}_N$ 1-form symmetry. Monte Carlo studies show that the self-dual point is a first order phase transition for $N \leq 4$ [127]. When $N \geq 5$, the theory flows to gapless Maxwell theory at the coupling $e^2 = 2\pi/N$, where the coupling is fixed by matching the duality defect in the renormalization group flow using the non-invertible symmetry in Maxwell theory [28].

For general $N, p$, the dynamics of the lattice model is constrained by the non-invertible defect: it cannot flow to gapped phase preserving the non-invertible symmetry unless $p^2 = -1$ mod $N$. This indicates at these special values of $N, p$, there should be duality-symmetric terms that one can add to the Hamiltonian to make it gapped (at least in the bulk) at $J_0 = J_1$. We discuss this point more in Section 5.

### 4.3.1 Large $N$ limit

For $p = 0$, for sufficiently large $N$, the lattice model flows to gapless Maxwell theory at coupling $\tau_{U(1)} = iN$, where the coupling is obtained by matching the non-invertible symmetry [39]. Here we make some conjectures of what might happen at nonzero $p$.

Let us impose the vertex term of $H_1$ in Figure 3 exactly to enforce the Gauss law, and we express the operators $Z$ and $X$ using the $U(1)$ gauge field $a$ and its canonical electric field $\Pi$ (we choose the temporal gauge $a_0 = 0$) as

$$Z = e^{ia_j}, \quad X = e^{\frac{2\pi i}{N}\Pi_j} , \tag{4.4}$$

where $j$ labels the direction of the edge, and we used $[\Pi, a] = 1$. In the Euclidean spacetime picture, the terms in the first row in Figure 3 are small loop operators with two edges in the temporal direction. In the continuum limit, they contribute the action $\int (\frac{2\pi}{N}\Pi_i + ip\epsilon_{0ijk}B_{jk})^2$, where we denote the magentic field as $B_{ij} = \partial_i a_j - \partial_j a_i$. The terms in the last line of Figure 3 are small loop operators in spatial directions, and in the continuum limit they contribute the action $\int (B_{ij} + \frac{2\pi p}{N}i\epsilon_{0ijk}\Pi_k)^2$. Thus for $N$ sufficiently large compared to $p$, the theory at low energy is described by the free $U(1)$ gauge theory. When $p$ is larger, the quadratic terms above are not sufficient to capture the Hamiltonian terms, and we expect the theory is not the Maxwell theory, as discussed further below.

**Domain wall tension** Consider the interface where on half space we perform the $S$ gauging (with the appropriate shift and $X \leftrightarrow Z$). Then along the interface, there is additional energy cost from the terms in $H_0$ and $H_1$, because the terms do not commute. For $p < 2\sqrt{N}$, such energy cost can be estimated, for $N \gg 1$, as $1 - e^{2\pi i(p^2)/N} \propto \sin\left(\frac{2\pi(p^2)}{N}\right)$, from the commutation relation of $Z^p, X^p$ and $Z, X$.

From the domain wall tension, we expect the following

- When $p^2 \ll N$, the tension vanishes for large $N$, and the duality symmetry is unbroken. For large $N$ the phase is described by gapless Maxwell theory.

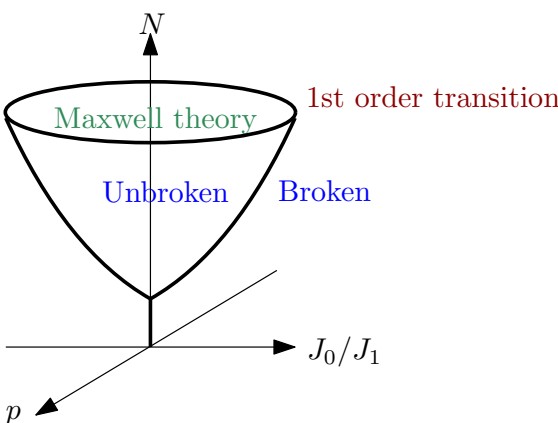

Figure 6: Proposed phase diagram for the lattice model: inside the cone the duality symmetry is unbroken but the theory is gapless, while outside the cone the symmetry is broken. At the upper part of the interior of the cone for large $N$ and small $p$, the phase is described by gapless Maxwell theory. The shell of the cone are first order phase transitions. The $N, p$ plane through the origin corresponds to $J_0 = J_1$.

- When $\frac{p^2}{N} \sim 1/4$, the tension is finite for large $N$, and the duality symmetry is broken.

Since $N, p$ are discrete, we do not expect new fixed points from dialing $p/N$. Therefore we propose that the phase diagram in terms of the three parameters $p, N$ and the lattice coupling $J_0/J_1$ is given by cone extending from small $p, N$ and $J_0 = J_1$, where inside the cone the duality symmetry is unbroken, and for large $N$ the upper part of the cone is the Maxwell theory. The plane at $p = 0$ should match with the phase diagram of Ref [65] (see also [127]). Outside the cone, the duality symmetry is broken. The boundary cone represent first order phase transitions. See Figure 6 for an illustration.

## 4.4   $ST^n$ gauging

The action of $ST^n$ gauging on the lattice is more subtle, when $\gcd(n, N) = 1$. This is because there are different versions of the $\mathbb{Z}_N$ 1-form SPT given by $H_0$ and $H_1$, that may differ by a gravitational term. $ST^n$ gauging for $\gcd(n, N) = 1$ can be implemented as $S \circ U_0$ or $S \circ U_1$, where $U_0$ and $U_1$ are the unitary operators that entangle the $H_0$ 1-form SPT or the $H_1$ 1-form SPT respectively. Specifically, $U_0^\dagger H_0^{(0)} U_0 = H_0^{(n)}$ and $U_1^\dagger H_0^{(0)} U_1 = H_1^{(n)}$, where the superscript denotes the $\mathbb{Z}_N$ 1-form SPT class (i.e. $p = n$). $U_0$ is a finite-depth quantum circuit, while $U_1$ may be a QCA due to the nontrivial surface topological order with $c \neq 0$ [157]. We can equivalently implement the $ST^n$ gauging using modified Gauss laws. We illustrate the modified Gauss laws corresponding to $S \circ U_0$ in Figure 4. For $\gcd(n, N) = 1$, there is an alternate set of modified Gauss laws corresponding to $S \circ U_1$.

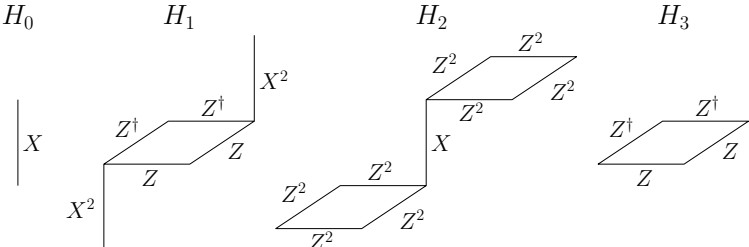

Figure 7: $ST^4$ cyclically maps the trivial paramagnet $H_0$ to the twisted deconfined gauge theory $H_1$. It them maps the gauge theory to the class $p = 4$ SPT $H_2$ and finally to the untwisted deconfined gauge theory $H_3$. We have only shown one of the three kinds of terms (and neglected the vertex terms) of each Hamiltonian.

Due to this ambiguity, we will discuss in this section an example where $\gcd(n, N) \neq 1$. [20] We will give a lattice model similar to the ones above, but for $ST^4$ gauging, and we use as a particular example $N = 4$. Because $ST^4$ is not order 2 (up to charge conjugation, which leaves the SPT invariant), we actually obtain a model with an $ST^4$ invariant multicritical point. Specifically, the model takes the form

$$H = J_0 H_0 + J_1 H_1 + J_2 H_2 + J_3 H_3, \tag{4.5}$$

where the Hamiltonian terms for $H_0, H_1, H_2$, and $H_3$ are illustrated in Figure 7. The order four $ST^4$ symmetry cyclically permutes $H_0, H_1, H_2$, and $H_3$, so at the point $J_0 = J_1 = J_2 = J_3$, the theory is invariant under $ST^4$.

## 4.5 Fusion rule and condensation defect

Let us study the $\bar{\mathcal{D}} \times \mathcal{D}$ fusion rule of the non-invertible $S$ symmetry. We will insert the symmetry defect by gauging on part of space, instead of on the entire space, with the rough boundary condition (*i.e.* Dirichlet boundary condition), by setting $\tilde{Z} = 1$ on the plaquettes on the domain wall. We consider the convention of starting with the gauge theory in all of space. Let us take the defect $\bar{\mathcal{D}}$ to be at coordinate $z = -\epsilon$ and the defect $\mathcal{D}$ to be at $z = \epsilon$. We then shrink the strip $-\epsilon < z < \epsilon$.

We remove the second and third "windmill" terms with $\tilde{X}, \tilde{X}^\dagger$ in the first line of Figure 5, since they have plaquettes with $\tilde{X}, \tilde{X}^\dagger$ on the domain wall, and thus they do not commute with $\tilde{Z} = 1$ on the domain wall plaquettes.

The remaining second "windmill term" in the first line of Figure 5 becomes a product of four $\tilde{X}$ (or $\tilde{X}^\dagger$ depending on the orientation) on the edges of a cross on the domain wall, where the edge variables are labelled by the variables on the plaquettes that end on the edges on the domain wall. See the first term in Figure 8.

---

[20]We plan to study the subtleties of general $ST^n$ gauging in forthcoming work.

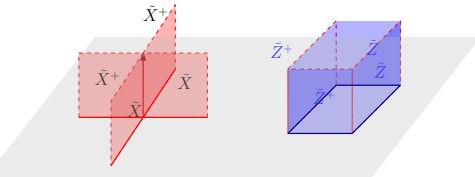

Figure 8: Condensation defect on the domain wall (grey shade) given by $\mathbb{Z}_N$ toric code model in 2+1d, where the variables on the domain wall edges (dark red and dark blue) are labelled by the variables on the plaquettes ending on the domain wall.

Similarly, the cube terms in the second line of Figure 5 becomes a product of four $\tilde{Z}$ (or $\tilde{Z}^\dagger$ depending on the orientation) on the edges around a plaquette on the domain wall, see the second term in Figure 8. Those are the standard vertex term and plaquette term in $\mathbb{Z}_N$ toric code model in 2+1d [158]. Thus we find that

$$\overline{\mathcal{D}} \times \mathcal{D} = \text{Condensation Defect} , \tag{4.6}$$

where the condensation defect is described by $\mathbb{Z}_N$ gauge theory on the domain wall. This reproduces the fusion rule in *e.g.* [28] for $\bar{\mathcal{D}} \times \mathcal{D}$.

# 5 Outlook

In this work we presented a general framework for studying anomalies of non-invertible symmetries in 3+1d, using a corresponding 4+1d Abelian 2-form gauge theory with a 0-form symmetry. We found that for the particular example of Kramers-Wannier like symmetries, certain symmetries are anomalous due to a quantity that generalizes the 2+1d FS indicator. These provide anomalies beyond those studied in Refs. [28, 39, 65]. We also uncovered a family of non-invertible time-reversal symmetries in 2+1d and presented some lattice models with these non-invertible symmetries.

Let us comment on some future directions which we would like to revisit:

- It would be interesting to study how non-invertible symmetries might fractionalize in 3+1d. This was studied in depth in *e.g.* [113, 144, 159, 160] for invertible symmetries.

- We can generalize to study topological defects of other dimensions including those related to continuous symmetries. As a particular example, the anomaly field theory that describes mixed anomaly of $U(1)$ higher form symmetries with is given by the analogue of Chern-Simons term for the gauge fields. It would be interesting to study the possible gapped or gapless boundaries of such examples from this point of view.

- In principle, our discussion does not need to assume Poincaré invariance, and should apply also to condensed matter systems. For instance, the anomaly field theory is Witt

equivalent to the $\mathbb{Z}_N$ two-form gauge theory described by the loop toric code model in 4+1d. This is unlike the argument in [65]. It would be interesting to apply our results to a more broad class of systems with less spacetime symmetry.

- The discussion can be generalized to defects that generate gauging a subsystem symmetry. See *e.g.* [161, 162] for examples of non-invertible subsystem symmetries.

- A natural future direction is to study other non-invertible symmetries in 3+1d by using different 0-form symmetry enriched 2-form Abelian gauge theories in 4+1d. For example, we can study anomalies of $ST^n$ symmetries in 3+1d. We already studied the first level obstructions in Section 2.2, but we did not systematically study anomalies from FS indicators. We did give an example in Section 3.5.1 showing that $ST^n$ symmetry may have obstructions to symmetric TQFTs related to beyond-cohomology SPTs not encountered for $S$ symmetry.

- The 0-form symmetry enriching the Abelian gauge theory can also be non-Abelian. For example, it can be a permutation group. It would be interesting to study anomalies of symmetries in 3+1d related to more general 0-form symmetries of the 4+1d Abelian gauge theory.

- In Section 3.6.2, we uncovered an infinite family of theories with non-invertible time reversal, originating from theories where the time-reversal had a mixed anomaly with charge conjugation. This method of obtaining a non-invertible time reversal symmetry from an anomalous time reversal symmetry is very general; we will discuss this and present many more examples in forthcoming work. [154]

- It would be very interesting to find realistic examples (beyond stacking constructions) where the FS indicators $\omega, \omega_f$ are nontrivial. It should also be possible to construct lattice models where $\omega$ is nontrivial. The lattice models we presented, like the Ising model, have trivial $\omega$. Furthermore, we restricted to bosonic models, but it should also be possible to construct fermionic models, especially with nontrivial $\omega_f$, for $N$ even.

- As mentioned in Section 4.3, the lattice models we constructed have predictable dynamics for $p = 0$. However, for $p \neq 0$, and in particular for $p \sim \sqrt{N}$, our models for large $N$ seem have very different dynamics. It would be interesting to study the phase diagram numerically, to confirm our hypothesized phase diagram in Figure 6. It would also be interesting to study the phase diagram for $p^2 = -1 \mod N$, where the bulk does not need to undergo a phase transition.

- If the phase diagrams for $N, p$ where $p^2 = -1 \mod N$ demonstrate that the $J_0 = J_1$ point is gapless, it would be interesting to study what symmetric perturbations can be used to take the system into a trivially gapped phase.

- There are many remaining questions related to the subtleties of gauging 1-form symmetries on the lattice. This is especially true for $ST^n$ gauging (see Section 4.4). We plan

to study the gauging, and recover the fusion rules (modified by translations) in future work.

## Acknowledgements

The authors thank Xie Chen, Meng Cheng, Anton Kapustin, Ho Tat Lam, Michael Levin, and Shu-Heng Shao for discussions. The authors thank Meng Cheng, Shu-Heng Shao, and Ryan Thorngren for comments on a draft. P.-S.H. is supported by Simons Collaboration of Global Categorical Symmetries. C.Z. is supported by the University of Chicago Bloomenthal Fellowship and the National Science Foundation Graduate Research Fellowship under Grant No. 1746045, and the Harvard Society of Fellows. C.C. is supported by the US Department of Energy Early Career program, the Sloan Foundation, and the Simons Collaboration on Global Categorical Symmetries. This work was performed in part at the Aspen Center for Physics, which is supported by National Science Foundation grant PHY-2210452. P-S.H. thanks Nordic Institute for Theoretical Physics for hosting the workshop Categorical aspects of symmetries.

## A  Non-invertible fusion rules from $\mathrm{Aut}(K)$ symmetry

When the domain wall that generates the bulk invertible $\mathrm{Aut}(K)$ symmetry ends on the boundary, it gives non-invertible symmetry on the boundary. Alternatively, if we gauge the bulk $g \in \mathrm{Aut}(K)$ symmetry, there is codimension operator given by open domain wall (such operator belongs to the twisted sector), and it generates new non-invertible symmetry.

We can compute the non-invertible fusion rules as follows. First, we obtain the worldvolume description of the domain wall using the action with properly quantized matrix $m$ for each term, and perform the $g$ transformation on $b^I$ on half spacetime $Y$ with boundary $\partial Y$ that supports the domain wall:

$$\frac{1}{2\pi} \sum_{I<J} \int_Y \left( (g^T m g)_{IJ} b^I db^J - m_{IJ} b^I db^J \right) = \frac{1}{4\pi} \sum_{I,J} \int_{\partial Y} W(g)_{IJ} b^I b^J \,, \qquad (A.1)$$

where the right hand side is the worldvolume action on the domain wall in the bulk, and $W(g)$ is a symmetric integer matrix that describes which excitations are condensed on the worldvolume of the $g$ domain wall. Explicitly,

$$W_{IJ} = \begin{cases} \sum_{K,L} g_{IK} m_{KL} g_{LJ} & I = J \\ \sum_{K,L} g_{IK} m_{KL} g_{LJ} - m_{IJ} & I \neq J \end{cases} . \qquad (A.2)$$

For instance, when the theory is $\frac{N}{2\pi} \int_{5d} b_e db_m$, under $g = T$ transformation $b_m \to b_m + b_e$, the $W$ matrix is given by $\frac{N}{4\pi} \int_{4d} b_e b_e$ [29]. Under $T = S$ transformation $(b_e, b_m) \to (b_m, -b_e)$, the

$W$ matrix is given by $\frac{N}{2\pi} \int_{4d} b_e b_m$ [29]. The domain wall itself is condensation with charges $ge_i - e_i$ where $e_i$ is the charge carried by the operator $e^{i \oint b^i}$ [49].

If we gauge the $g$ symmetry in the bulk, the open domain wall gives a codimension-two operator on its boundary which obeys non-invertible fusion rule. Denote the support of the domain wall by $M_4$ with boundary $M_3 = \partial M_4$, the fusion of $g$ with $\bar{g}$ can be obtained from $W(g)$ by the methods in *e.g.* [26, 28, 29, 35, 39, 49, 60]:

$$\partial M_4 = M_3 : \quad g(M_4) \times \overline{g(M_4)} = \# \sum_\lambda e^{i \sum_{I \leq J} W(g)_{IJ} \int_{\mathrm{PD}(\lambda^I/2\pi)} b^J + \frac{i}{4\pi} \sum_{I,J} \int_{M_3} \lambda^I d\lambda^J} \quad , \qquad (A.3)$$

where $\#$ is a normalization factor, and $\lambda = \{\lambda^I\}$ are one-forms on $M$ that take values in $\mathcal{A} = \mathbb{Z}^{2r}/K\mathbb{Z}^{2r}$, PD denotes the Poincaré dual on $M_3$.

Then the fusion rule in the absolute boundary theory with polarization $\mathcal{P}$ is given by further imposing the equivalence relation that sets the surface operators in $\mathcal{P}$ to be trivial.

We remark that the absolute theories for different polarizations can have different symmetries as in the examples in *e.g.* [104, 163].

## A.1 Example: $\mathbb{Z}_N$ two-form gauge theory

Let us consider $\mathbb{Z}_N$ two-form gauge theory

$$\frac{N}{2\pi} b_e db_m \; . \qquad (A.4)$$

We will derive the fusion rule for the non-invertible symmetry from the $ST^n$ transformation. The transformation is generated by the bulk domain wall

$$\int_{4d} \left( \frac{N}{2\pi} b_e b_m + \frac{nN}{4\pi} b_m b_m \right) \; . \qquad (A.5)$$

The fusion rule for defect ending on the boundary is given by

$$\mathcal{N} \times \overline{\mathcal{N}} = \sum_{S_e, S_m} e^{i \int_{nS_m + S_e} b_m + \int_{S_m} b_e} \; . \qquad (A.6)$$

On gapped boundary that preserves the $ST^n$ symmetry, where $N = \alpha^2 + \beta^2 + n\alpha\beta$, we can choose polarizations such as the Dirichlet boundary condition $b_e| = 0$ to obtain an absolute gapped boundary theory. For this polarization, the fusion rule is

$$\mathcal{N} \times \overline{\mathcal{N}} = \sum_{S_e, S_m} e^{i \int_{nS_m + S_e} b_m} \; . \qquad (A.7)$$

# B  Generalized Frobenius Schur-indicator and statistics

In this appendix, we discuss a similar obstruction to symmetric gapped phase as in [97] for topological lines in 1+1d, to volume operators in 3+1d. In the Abelian theory (2.14), suppose we consider an ordinary symmetry of order 2. We can obtain the anomaly field theory by gauging the symmetry. We have choice of gauging the symmetry with additional local counterterm. Suppose we add additional bosonic 4+1d Dijkgraaf-Witten theory for $\mathbb{Z}_2$ gauge group described by $H^5(B\mathbb{Z}_2, U(1)) = \mathbb{Z}_2$, the F symbol from fusing 5 volume operators give $(-1)$ since the Dijkgraaf-Witten term describes the 0-form symmetry anomaly [111, 164]. We will show that the volume operator in this case has "self-statistics" $i$ compared to the case without the Dijkgraaf-Witten term. This is similar to the 2+1d $\mathbb{Z}_2$ Dijkgraaf-Witten theory, where the $F$ symbol for fusing three vortex operators is the Frobenius–Schur indicator (see *e.g.* [116]).

More generally, let us consider Dijkgraaf-Witten theory for $\mathbb{Z}_2$ gauge group in odd spacetime dimension $(2n + 1)$. We can describe the theory by the action in continuous notation

$$\int_{M_{2n+1}} \left( \pi \frac{a}{\pi} \left( \frac{da}{2\pi} \right)^n + \frac{2}{2\pi} a db \right) \ , \tag{B.1}$$

where $a$ is a one-form and $b$ is a $(2n - 1)$-form. The equation of motion for $a$ gives

$$\frac{n+1}{(2\pi)^n} (da)^n + \frac{2}{2\pi} db = 0 \ . \tag{B.2}$$

Consider the correlation function on $M_{2n+1} = S^{2n+1}$ for the gauge invariant operator

$$\exp \left( i \oint_{V_{2n-1}} b + \frac{(n+1)i}{2(2\pi)^{n-1}} \int_{V_{2n}} (da)^n \right) \ , \tag{B.3}$$

where $V_{2n-1} = \partial V_{2n}$. With such operation insertion, the equation of motion for $b$ gives

$$da = -\pi \delta(V_{2n-1})^\perp, \quad a = -\pi \delta(V_{2n})^\perp \ , \tag{B.4}$$

where $\delta(V_{2n-1})^\perp$ is the delta function two-form that restricts to $V_{2n-1}$, and $\delta(V_{2n})^\perp$ is the delta function one-form that restricts to $V_{2n}$. Thus the correlation function is

$$\langle \exp \left( i \oint_{V_{2n-1}} b + \frac{(n+1)i}{2(2\pi)^{n-1}} \int_{V_{2n}} (da)^n \right) \rangle$$
$$= \exp \left( (-1)^{n+1} \frac{\pi i}{2^n} \int_{M_{2n+1}} \delta(V_{2n})^\perp (\delta(V_{2n-1})^\perp)^n + \frac{(-1)^n \pi (n+1)i}{2^n} \int_{M_{2n+1}} \delta(V_{2n})^\perp (\delta(V_{2n-1})^\perp)^n \right)$$
$$= \exp \left( (-1)^n \frac{ni\pi}{2^n} \int_{M_{2n+1}} \delta(V_{2n})^\perp (\delta(V_{2n-1})^\perp)^n \right) \ , \tag{B.5}$$

where the integral in the last line equals the intersection number of $V_{2n}$ and the one-dimensional $n$th intersection of $V_{2n-1}$. Thus nontrivial F symbol implies nontrivial self-statistics of the

magnetic operator. We can call the F symbol as generalization of the Frobenius-Schur indicator.

Let us consider some cases with lower $n$:

- For $4 + 1d$ spacetime dimension, $n = 2$, and the self statistics is $e^{i\frac{\pi}{2}} = i$. Thus we find that non-trivial F symbol (equals $(-1)$) for fusing 5 operators, which gives the Dijkgraaf-Witten term, implies nontrivial self-statistics.

- Similarly, for $2 + 1d$ spacetime dimension, $n = 1$, and the self-statistics is $e^{-\frac{\pi i}{2}} = -i$, as expected from the vortex in the $\mathbb{Z}_2$ Dijkraaf-Witten theory in 2+1d. The $F$ symbol (equals $(-1)$) for fusing three operators, which gives the Dijkgraaf-Witten term, implies nontrivial self-statistics.

# C   Trivializing the anomaly by symmetry extension

Consider the non-invertible symmetry associated with the $S$-transformation in bulk $\mathbb{Z}_N$, with $N > 2$. The $S$ transformation has order 4, and we can stack the bulk TQFT with an SPT phase for $\mathbb{Z}_4$ symmetry, described by

$$\frac{2\pi p}{4} \int A \frac{dA}{4} \frac{dA}{4} \ , \tag{C.1}$$

where $p$ is an integer, and $A$ is the background gauge field for the $\mathbb{Z}_4$ symmetry. This implies that the domain wall that generates the unit transformation is attached to the SPT phase

$$\frac{2\pi p}{4} \int \frac{dA}{4} \frac{dA}{4} \ . \tag{C.2}$$

We can trivialize the anomaly by decorating the domain wall with TQFTs to cancel such phase. For instance, we can extend the $\mathbb{Z}_4$ symmetry in the following way:

- We can extend $\mathbb{Z}_4$ to $\mathbb{Z}_{16}$, denote a lift of $\mathbb{Z}_4$ gauge field to $\mathbb{Z}_{16}$ as $\tilde{A}$ a $\mathbb{Z}_{16}$ cochain, then we can cancel the domain wall anomaly be decorating it with the Chern-Simons term

$$\frac{2\pi p}{16} \int \tilde{A} \frac{dA}{4} \ . \tag{C.3}$$

  The fourth power of the domain wall gives

$$\frac{2\pi p}{4} \int A \frac{dA}{4} \ . \tag{C.4}$$

  Thus the decorated domain wall generates symmetry $\mathbb{Z}_{16/\gcd(p,4)}$.

- We can extend $\mathbb{Z}_4$ symmetry to a three-group by decorating the domain wall with $\mathbb{Z}_4$ three-form gauge field that satisfies

$$dB_3 = p \frac{dA}{4} \frac{dA}{4} \ . \tag{C.5}$$

- We can decorate the domain wall with the gapped boundary of the invertible TQFT $\mathbb{Z}_4 \times \mathbb{Z}_4$ two-form gauge theory with the topological action $\frac{2\pi p}{4} \int b \cup b'$ for the two-form gauge fields $b, b'$,[21] such that the transformations of $b, b'$ are correlated with that of $dA/4$. Encircling the surface operators $\int b, \oint b'$ on the junction of the domain wall such that $dA/4$ is non-trivial picks up a factor of $i$. Then the domain wall becomes non-invertible:

$$\mathcal{N} \times \overline{\mathcal{N}} = \sum_{S,S'} e^{\frac{2\pi i}{4} \int_S b + \frac{2\pi i}{4} \int_{S'} b'} , \tag{C.6}$$

where we omit an overall normalization factor on the right hand side.

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
