# Peer review of "Anomalies of Non-Invertible Symmetries in (3+1)d"

_SciPost Physics_

## Round 2 · Referee Report · Anonymous (Referee 1) · 2024-5-14

Strengths

1 - The paper studies anomalies for non-invertible symmetries in 3+1d. The authors identify two levels of anomalies, generalizing analogous results for anomalies of Tambara-Yamagami non-invertible symmetries in 1+1d
2 - Along the way, the authors study anomalies in the time reversal symmetry of an interesting class of 2+1d TQFTs, and construct an infinite family of 2+1d TQFTs enjoying a non-invertible version of time reversal symmetry
3 - The authors present and study explicit lattice models for 3+1d theories that enjoy a ZN 1-form symmetry and are invariant under its gauging

Weaknesses

1 - At times, the discussion is rather technical, especially in section 3

Report

Uncovering dynamical consequences of non-invertible symmetries in 3+1d dimensions is an important endeavor. This paper makes substantial progress in this direction by performing a systematic analysis of 't Hooft anomalies for Kramers-Wannier-like symmetries.

This paper meets this Journals' acceptance criteria and is recommended for publication, after addressing some minor points listed below.

Requested changes

1 - The notation GL(n,Z) is often used to indicate n x n matrices with integer entries that are invertible and whose inverses are also matrices with integer entries. Thus, these matrices have determinant equal to +1, -1. The authors use the notation GL(2r,Z) around (2.8), but remark that U needs not have determinant +1 or -1. The authors could comment briefly on this for clarification purposes.
2 - Some minor typos: "that the the Lagrangian" beginning of sec 2.1.2; "an quadratic" above (2.12); "an symmetric" above (2.24); "affect out discussion" below (3.19); "odd N cause" 4th line page 29; "symmetry symmetry" between (3.42) and (3.43); "can can produce" above Thm 2; "J0 and and J1" bottom page 36; "the anomaly field theory that [...] form symmetries with is given by" 2nd bullet page 40

Recommendation

Ask for minor revision

---

## Round 2 · Referee Report · Anonymous (Referee 2) · 2024-8-25

Report

This work provides a new way to determine the anomaly of non-invertible symmetry in higher dimensions. The key idea is to consider the abelian symmetry TQFT associated with the abelian part of the symmetry, and examine its topological boundary conditions.

The first level of anomaly comes from the non-existence of duality invariant topological boundary condition of the abelian symmetry TQFT.

The second level of anomaly comes from the FS indicator, which, by decorated domain wall construction, reduces to the anomaly of time reversal symmetry of the non-invertible defect worldvolume theory. Then (1) the non-invertible symmetry being anomaly free (in the sense of admitting a trivially gapped phase) is argued to be equivalent to (2) the existence of a 3d world volume TQFT that matches the time reversal anomaly from the FS indicator. This equivalence between (1) and (2) are scattered in different parts of the paper, and I would suggest the authors to establish the equivalence better by, e.g. having a dedicated subsection, and explain the proof of (1) -> (2) and (2) -> (1) directions explicitly, therefore showing their equivalence. I believe this would make the paper, especially Section 3, more readable.

Apart from the above suggestion, there is a small gap for the argument around eq 2.2. One needs to argue that simple surface induces only 1 copy of identity line. Otherwise eq 2.2 can be modified to $n \times n’ = \sum_i n_i$, which does not lead to contradiction.

Recommendation

Ask for minor revision

---

## Editorial Decision

resubmitted